# Diffusion Models are Secretly Zero-Shot 3DGS Harmonizers

**Vsevolod Skorokhodov**[*]                                   *vsevolod.skorokhodov@epfl.ch*
*Computer Vision Laboratory, EPFL*

**Nikita Durasov**[*]                                          *nikita.durasov@epfl.ch*
*Computer Vision Laboratory, EPFL*

**Pascal Fua**                                                *pascal.fua@epfl.ch*
*Computer Vision Laboratory, EPFL*

**Reviewed on OpenReview:** `https://openreview.net/forum?id=1jjIitxVmM`

## Abstract

Gaussian Splatting has become a popular technique for various 3D Computer Vision tasks, including novel view synthesis, scene reconstruction, and dynamic scene rendering. However, the challenge of natural-looking object insertion, where the object's appearance seamlessly matches the scene, remains unsolved. In this work, we propose a method, dubbed D3DR, for inserting a 3DGS-parametrized object into a 3DGS scene while correcting its lighting, shadows, and other visual artifacts to ensure consistency. We reveal a hidden ability of diffusion models trained on large real-world datasets to implicitly understand correct scene lighting, and leverage it in our pipeline. After inserting the object, we optimize a diffusion-based Delta Denoising Score (DDS)-inspired objective to adjust its 3D Gaussian parameters for proper lighting correction. We introduce a novel diffusion personalization technique that preserves object geometry and texture across diverse lighting conditions, and utilize it to achieve consistent identity matching between original and inserted objects. Finally, we demonstrate the effectiveness of the method by comparing it to existing approaches, achieving 2.0 dB PSNR improvements in relighting quality.

openreview / code / web

## 1 Introduction

3D object insertion is a computer vision problem that arises when placing a 3D object from one scene into a specific location in another. The task is to adjust the appearance of the object to ensure consistency with the lighting of the new scene, making the insertion appear realistic. Traditional methods rely on physically-based rendering, which requires complex modeling and manual parameter tuning of scene properties such as texture, material, and lighting. This process is often slow, labor intensive, and prone to inaccuracies, since precisely reconstructing the illumination and reflectance of a real scene remains highly challenging.

Recent advances in novel view synthesis (NVS), and in particular the emergence of 3D Gaussian Splatting (3DGS) (Kerbl et al., 2023), have revolutionized the way scenes are represented in computer vision. However, a key challenge when inserting an object into a scene — specifically that the lighting of the inserted object does not match the lighting of the scene, making the insertion appear unrealistic — remains unresolved for 3DGS representation. Although several approaches have been proposed for other parametrizations (Li et al., 2020; Song et al., 2022; Liang et al., 2024; Ye et al., 2024; Jin et al., 2023; Zhang et al., 2021), they are either not directly applicable to 3DGS or yield unsatisfactory or unrealistic results, highlighting the need for new methods better suited to the 3DGS object insertion task.

In recent years, the scientific community has discovered hidden zero-shot capabilities of diffusion models, such as image classification (Li et al., 2023) and cartoon-style image creation (Zhao et al., 2023). In this work, we

---

[*]Equal contribution.

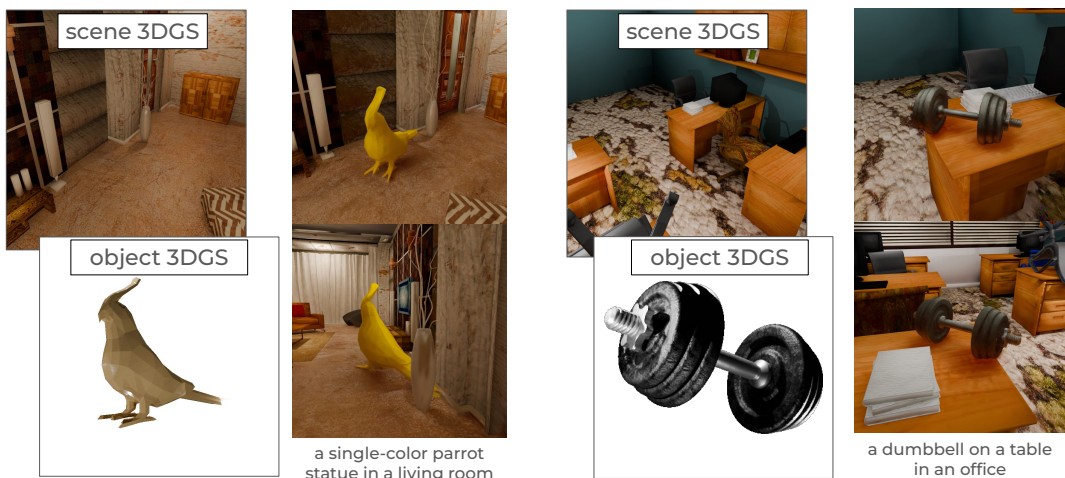

Figure 1: **Overview of the task.** Our method aims to insert a 3DGS object into a specific location in a 3DGS scene, followed by adjusting the object's appearance to match the scene's lighting. The final result is also a new 3DGS scene that includes both the input scene and the object with realistic lighting.

present a novel zero-shot scenario of diffusion models — lighting harmonization during 3D object insertion. More specifically, we show that the 3DGS representation can leverage this hidden ability of diffusion models to enforce realistic lighting consistency between inserted objects and their environments.

Following modern image-to-3D pipelines (Zhuang et al., 2024; Raj et al., 2023; Huang et al., 2023), we employ DreamBooth (Ruiz et al., 2023) for object-specific adaptation. However, DreamBooth frequently fails to reconstruct fine details (e.g., small prints, decorative patterns) (He et al., 2025), which are critical for preserving object identity. We therefore introduce a personalization technique that enhances fine-detail fidelity by conditioning a diffusion model on the original 3DGS object renderings.

As shown in Fig. 2, we first personalize diffusion models on the target object, insert it into a new scene, and then refine its appearance by optimizing a diffusion-based Delta Denoising Score (DDS) objective (Hertz et al., 2023). This objective enforces consistency with the surrounding scene while preserving structural detail. By using diffusion models trained on large-scale image data, our approach enables effective object relighting without the need for environmental maps or complex material estimation, outperforming existing methods in lighting quality and efficiency for realistic 3D object insertion.

Thus, our main contributions are as follows:

- We show that diffusion models possess an inherent ability to perform 3DGS object insertion and relighting without any supervised training data, enabling a fully zero-shot approach.

- We introduce a diffusion-based 3DGS object insertion and relighting method that enforces lighting consistency when integrating 3D objects into 3DGS scenes.

- We propose a diffusion personalization approach designed for object texture details (such as prints, decorative patterns and etc.) preservation.

- We collect a diverse dataset of 3DGS scenes and objects and demonstrate through extensive benchmarking that our method significantly outperforms existing approaches in relighting quality.

## 2 Related Work

### 2.1 3D Gaussian Splatting

Neural Radiance Fields (NeRFs) (Mildenhall et al., 2021) have advanced novel view synthesis by modeling scenes as continuous volumetric representations using MLPs. However, NeRFs need costly optimization and

have slow rendering times. Gaussian Splatting (3DGS) (Kerbl et al., 2023) offers an efficient alternative by explicitly representing scenes with 3D Gaussians, enabling real-time rendering faster than the quickest NeRF (Müller et al., 2022) while preserving visual quality of state-of-the-art NeRFs (Barron et al., 2021).

In 3DGS, a scene is represented by a set of 3D Gaussians, where each Gaussian is characterized by its mean $\mu$, covariance $\Sigma$ (represented by a quaternion $\mathbf{q}$ and scaling vector $\mathbf{s}$), opacity $o_i$, and color information $\mathbf{c}_i$, which is view-dependent and modeled by spherical harmonics (SH). Rendering is performed by projecting Gaussians onto the image plane and compositing their contributions through alpha blending (Kerbl et al., 2023), a technique that combines semi-transparent elements based on their opacities, to produce the final pixel colors. In summary, a scene is represented as a collection of Gaussians, each with parameters $\{\mu_i, \mathbf{s}_i, \mathbf{q}_i, o_i, \mathbf{c}_i\}$.

Recent advancements in 3DGS have improved its scalability (Kerbl et al., 2024), deblurring (Zhao et al., 2024; Peng et al., 2024), handling of dynamic scenes (Wu et al., 2024), integration with diffusion-based generative models for 3D content creation (Tang et al., 2023), and scene editing (Chen et al., 2024). From the best of our knowledge this is the first work effectively addressing 3D object insertion problem for objects and scenes in the 3DGS representation through diffusion-based optimization.

## 2.2 Diffusion models

Diffusion models (Ho et al., 2020; Song et al., 2020) have achieved state-of-the-art generative performance in image synthesis (Ramesh et al., 2022), inpainting (Lugmayr et al., 2022), super-resolution (Li et al., 2022) and others. Diffusion models denoise an image step-by-step, from a pure gaussian noise to the desired image (Ho et al., 2020). Classifier-free guidance (Ho & Salimans, 2022) is usually applied for text-to-image generation, when the predicted noise is calculated using conditional and unconditional outputs:

$$\varepsilon_\phi^\omega(x; y; t) = \varepsilon_\phi(x; \varnothing; t) + \omega(\varepsilon_\phi(x; y; t) - \varepsilon_\phi(x; \varnothing; t)), \tag{1}$$

where $\varepsilon_\phi$ is a diffusion model, $x$ is a noisy image, $t$ is a timestep, $y$ is an object representative feature (e.g. text prompt), $\omega$ is a guidance scale.

In the context of 3D generation, diffusion models were successfully applied in DreamFusion (Poole et al., 2022), where NeRF generation is optimized by computing gradients through a diffusion process. This uses the Score Distillation Sampling (SDS) method, which derives gradients from the discrepancy between the noise predicted by a diffusion model and the actual noise added to a NeRF-rendered image at each timestep, and backpropagates them to update the NeRF's MLP parameters (Poole et al., 2022):

$$\nabla_\theta L_{sds} = \mathbb{E}_{t,\varepsilon}\left[\omega(t)\left(\varepsilon_\phi^\omega(\alpha_t g(\theta) + \sigma_t \varepsilon; y; t) - \varepsilon\right)\frac{\partial g(\theta)}{\partial \theta}\right], \tag{2}$$

where $\theta$ represents scene parameters (such as NeRF's MLPs), $g$ is a differentiable rendering function, $\omega(t)$, $\alpha_t$, and $\sigma_t$ are diffusion process parameters, and $\varepsilon$ is random Gaussian noise.

Delta Denoising Loss (DDS) (Hertz et al., 2023), an extension of the SDS loss originally introduced for image editing tasks, has also been applied to 3D editing (Chen et al., 2024). Addressing the limitation of SDS, which are blurry and oversaturated results, DDS reduces these effects (Hertz et al., 2023). The gradient of the DDS loss with respect to $\theta$ is defined in Eq. 3:

$$\nabla_\theta L_{dds} = \mathbb{E}[(\varepsilon_\phi^\omega(\alpha_t g(\theta) + \sigma_t \varepsilon; y; t) - \varepsilon_\phi^\omega(\alpha_t g(\theta_{orig}) + \sigma_t \varepsilon; y_{orig}; t))\frac{\partial g(\theta)}{\partial \theta}], \tag{3}$$

where $\theta$ and $\theta_{orig}$ denote the optimizable and original scene parameters, and $y$ and $y_{orig}$ are prompts describing the desired and original states, respectively. Initially, $\theta = \theta_{orig}$, and during optimization $\theta_{orig}$ remains fixed while only $\theta$ is updated.

**Improving realism in 3D generation.** SDEdit (Meng et al., 2021) is a diffusion-based image editing method. It first adds noise to an input image and then denoises it using a diffusion model guided by a text prompt. For example, to turn a dog into a cat, the dog image is noised and then denoised with the prompt «a photo of a cat». The approach also enables improving the realism in 3D generation (Zhuang et al., 2024; Tang et al., 2023). This is done by rendering the 3D scene from various views, applying SDEdit (Meng et al., 2021) to the rendered images, and optimizing the 3D scene parameters to match the edited images.

**Diffusion Models for Lighting Refinement.** During the denoising process, diffusion models transform Gaussian noise into a realistic image, gradually aligning it with the distribution of real images (Ho et al., 2020). Trained on large datasets of naturally lit images, diffusion models implicitly learn the underlying lighting characteristics. These learned priors can be utilized during the inpainting task, as done in Diffusion-Light (Phongthawee et al., 2024). The authors fine-tune a diffusion model on a set of inpainting examples and use it to inpaint a chrome ball at the image center, from which they infer environmental lighting. The resulting estimates are remarkably accurate, indicating that the model reproduces realistic illumination.

The SDS image generation process is conceptually similar to denoising. The initial image is pure Gaussian noise, which gradually becomes realistic through step-by-step optimization with the SDS loss (Poole et al., 2022). Consequently, we hypothesize that optimizing with SDS/DDS loss encourages images to match the lighting statistics of real-world photos, transferring learned priors to produce more natural and consistent illumination. We show that this hypothesis holds in Sec. 3.3, where we apply a similar idea for lighting enhancement as in DiffusionLight (Phongthawee et al., 2024), performing inpainting with the DDS loss.

Latent Bridge Matching (Chadebec et al., 2025) achieves state-of-the-art results in 2D object harmonization. The authors define the diffusion process as a mapping from images with copy-pasted objects to harmonized ones, instead of Gaussian noise to real images. However, the method fails to produce shadows and lacks view consistency when applied to 3D object insertion, as shown in our experiments (Sec. 4, App. 17).

## 2.3 3DGS Object Insertion

TIP-Editor (Zhuang et al., 2024) generates a 3DGS object from scratch at a specific location within a 3DGS scene, using a set of scene images, a single object picture, a bounding box, and prompts describing the object and the scene. It first personalizes a diffusion model on the scene and the object following DreamBooth (Ruiz et al., 2023), and then generates the object 3DGS from scratch using the SDS loss. The method faces difficulties in generating large objects, as further discussed in Sec. 4.3.

Relightable 3D Gaussians (R3DG) (Gao et al., 2025) extends classical 3D Gaussian Splatting by learning scene lighting and physically based per-Gaussian parameters such as albedo, roughness, etc. As a result, inserting a 3DGS object into another 3DGS scene becomes a straightforward copy-and-paste operation followed by physically based rendering. However, R3DG fails to properly reconstruct some essential object properties, such as albedo, because it lacks prior knowledge about the object and cannot distinguish between actual dark colors and shadows. Furthermore, the method models incident lighting as a combination of a global environmental light and Gaussian-specific indirect components, which depend on the original scene and thus become invalid when the object is placed elsewhere.

Instruct-GS2GS (Vachha & Haque) performs diffusion-based 3DGS editing using the 2D image editing model InstructPix2Pix (Brooks et al., 2023). It iteratively updates the training dataset by applying 2D edits to rendered images while optimizing the underlying 3DGS representation. However, relighting is challenging for InstructPix2Pix (Bashkirova et al., 2023) and the method fails to produce realistic object appearance.

## 2.4 3D Object Placement

Our method assumes a user-defined object location and focuses on realistic visual integration after insertion. Recent works such as PlaceIt3D (Abdelreheem et al., 2025) address language-guided 3D object placement by predicting plausible object locations in a scene. These methods solve the where-to-place problem, whereas we focus on the how-to-insert problem by ensuring photometric consistency and realistic shadows. The two directions are complementary and can be combined for fully automated object insertion.

# 3 Method

## 3.1 Overall Pipeline

In this section, we present our approach for inserting a 3D object into a 3D scene when both are represented by 3D Gaussians (3DGS). The method requires initial 3DGS models of the object and the scene, as

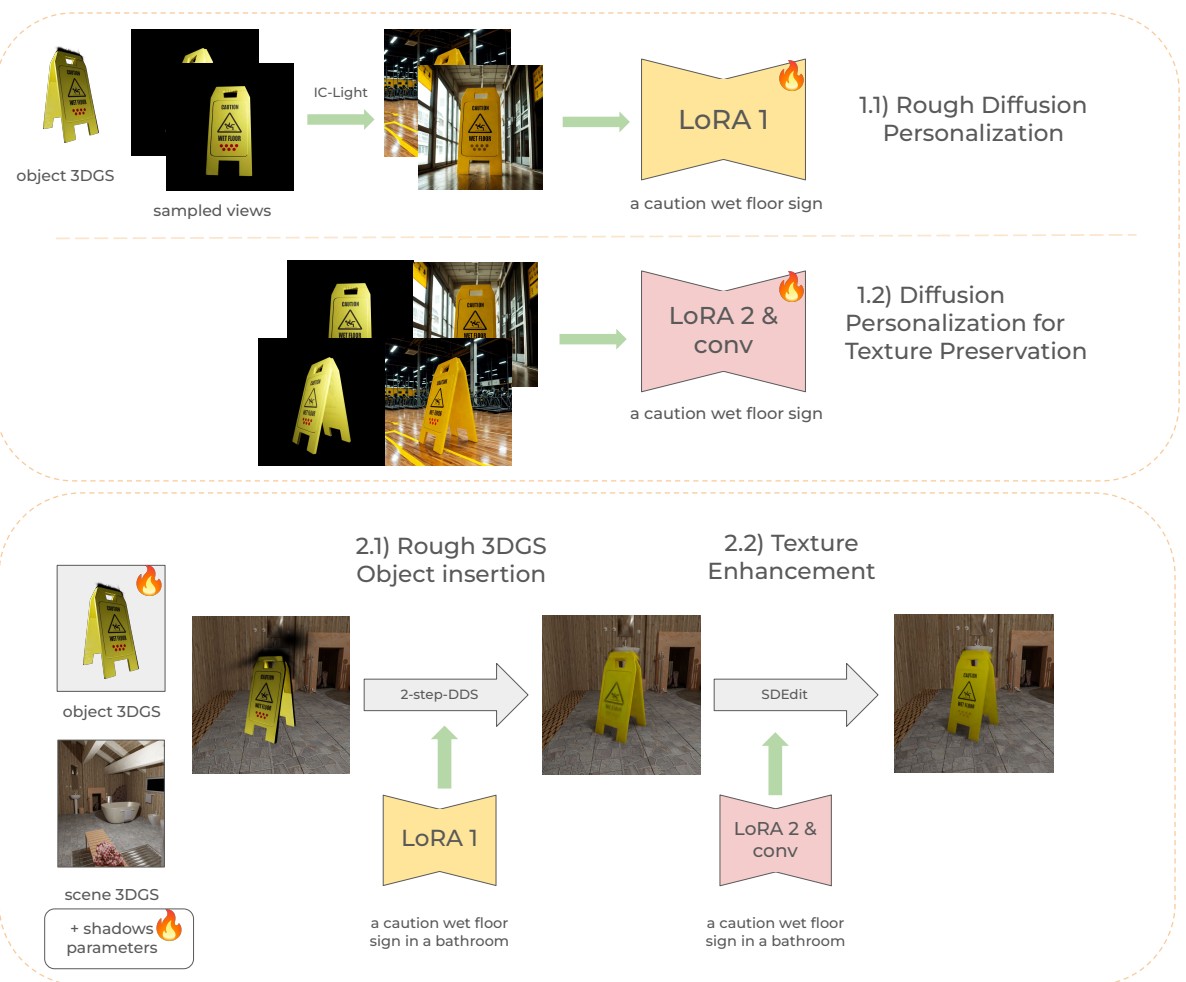

Figure 2: **Pipeline overview**. The method is able to perform 3D object insertion of a 3DGS object into a 3DGS scene with object light correction. The whole pipeline consists of two steps. 1) a diffusion model is personalized on the object. 2) 2-step-DDS is utilized to adjust the object appearance after 3DGS insertion. The flame icon denotes that the parameters are optimized (for LoRA during personalization, object and shadow parameters during 2-step-DDS).

well as prompts describing them for proper utilization of diffusion models. The approach outputs 3DGS representations of the object and the scene with corrected appearances after insertion.

An overview of our pipeline is provided in Fig. 2. The method comprises two main steps: 1) the *diffusion personalization step* (Sec. 3.2), which integrates object-specific information into two diffusion models for object structure preservation, and 2) the *3DGS insertion step* (Sec. 3.3), which leverages the personalized diffusion models from the first step to adjust the object and scene appearance, making the insertion look realistic. To achieve accurate relighting and avoid color artifacts that typically arise from naive DDS optimization, we introduce a modified version of the DDS loss described in Sec. 3.4. To further improve realism and texture preservation, we perform a refining phase using SDEdit (Meng et al., 2021) with a personalized diffusion for object texture preservation, which differs from the one used during DDS optimization.

## 3.2 Diffusion Personalization

We personalize diffusion models to preserve object identity — a necessary requirement for realistic 3DGS object insertion. To this end, we use two personalized diffusion models. The first is referred to as the

*rough personalization model* $\varepsilon_{\phi,r}$, which closely follows the DreamBooth (Ruiz et al., 2023) training pipeline. However, DreamBooth typically preserves only the main color and some texture details, often failing to fully maintain object identity and altering fine textures (He et al., 2025). To overcome this limitation, we develop a second personalized model referred to as the *texture-preserving personalization model* $\varepsilon_{\phi,t}$, whose architecture and training procedure are specifically designed to preserve object identity.

**Rough Diffusion Personalization** Let $\varepsilon_\phi$ denote a pretrained diffusion model. To train the *rough personalization model* $\varepsilon_{\phi,r}$ we follow the DreamBooth (Ruiz et al., 2023) for personalized image generation by fine-tuning a pretrained diffusion model $\varepsilon_\phi$ on a small set of object images $X^o = \{x_1^o, \ldots, x_N^o\}$. To prevent overfitting and maintain output diversity, DreamBooth uses *class preservation* — generating additional images $X^{class} = \{x_1^{class}, \ldots, x_k^{class}\}$ from the same class as the target object and adding them to the training set $X^o$. However, classical DreamBooth personalization tends to overfit to a single object appearance when all object images $X^o$ are captured under the same lighting, as discussed in Sec. 4.5.

To mitigate this issue, we utilize IC-Light (Zhang et al., 2025), which synthesizes object images under varied illumination and backgrounds. IC-Light takes an object image $x^o$, a descriptive object prompt, a background prompt, and a lighting direction as inputs. Our approach involves sampling $N$ random viewpoints to render object images $X^o$ via 3D Gaussian Splatting (3DGS), selecting $N$ random backgrounds from 20 predefined environments (e.g., *kitchen, shop, park, library*), and choosing lighting directions from four options (*right, left, up, down*). IC-Light then creates a diverse set of images $X^{ic} = \{x_1^{ic}, \ldots, x_{ic}^N\}$ under various lighting conditions where each relit image $x_i^{ic}$ is generated from the corresponding render $x_i^o$ using IC-Light. Finally, we fine-tune a Diffusion Model $\varepsilon_\phi$ with LoRA (Hu et al., 2022) using DreamBooth with images $\{X^{ic}, X^{class}\}$. More details on the personalization parameters are provided in Sec. 4.

**Personalization for Texture Preservation.** To train the *texture-preserving personalization model* $\varepsilon_{\phi,t}$ we design a novel diffusion personalization strategy. The idea is based on the fact that the original 3DGS object already contains the necessary texture information, which can be leveraged to guide personalized diffusion models toward preserving object textures during generation. Inspired by how InstructPix2Pix (Brooks et al., 2023) handles additional image information in a diffusion-based image editing framework, we similarly modify the first convolutional layer of the diffusion model to accept two inputs: (1) the object image $x^o$ under its original lighting condition and (2) the noisy image $x^{noisy}$ obtained by adding diffusion noise to the relit image $x^{ic}$ generated from $x^o$ using IC-Light.

Specifically, we add a convolutional branch that processes the original object image $x^o$, while the noisy image $x^{noisy}$ passes through the original branch. The outputs are summed before entering into the subsequent layers. The new branch is initialized with zero weights to preserve the original model $\varepsilon_\phi$ behavior at the start of training.

We incorporate LoRA (Hu et al., 2022) to allow the diffusion model $\varepsilon_\phi$ to effectively process the additional information introduced by the new convolutional layer. We fine-tune the modified diffusion model $\varepsilon_{\phi,t}$ using the original 3DGS object renderings $X^o = \{x_1^o, \ldots, x_N^o\}$ and their IC-Light-processed counterparts $X^{ic} = \{x_1^{ic}, \ldots, x_N^{ic}\}$. During training, only the weights of the first convolutional layer, the new convolutional layer, and the LoRA parameters are optimized. Generally, the proposed approach enables the model $\varepsilon_{\phi,t}$ to better capture the interaction between object appearance and the desired textures, with additional optimization details and results provided in App. K.

In our experiments in Sec. 4.5, we show that the rough personalization model $\varepsilon_{\phi,r}$ performs relighting effectively but lacks detailed object knowledge, preserving only the main color. The texture-preserving personalization model $\varepsilon_{\phi,t}$ produces realistic textures but does not achieve accurate relighting.

### 3.3 Delta Denoising Score (DDS)

As discussed in Sec. 2.2, DiffusionLight (Phongthawee et al., 2024) demonstrates that, after diffusion in-painting of a chrome ball into an image, the ball exhibits the correct appearance. We show that DDS-based image inpainting similarly corrects the object's lighting; however, it works only with proper initialization. To illustrate this idea, we use a toy dataset of cup images, described in detail in App. F.

We first demonstrate that DDS optimization *inherits* object appearance during inpainting. In the notation of Eq. 3, the reference image $g(\theta_{orig})$ is either a real image of a cup on a table (top left, Fig. 3) or another image of the same table with a copy-pasted cup from a different source (bottom left). We apply DDS to transform the cups into statue heads in both cases, initializing the optimized image $g(\theta) = g(\theta_{orig})$ and using prompts $y_{orig} =$ «*a cup on a plate*» and $y =$«*a statue head on a plate*». In this setup the DDS optimization is image editing, since the original image $g(\theta_{orig})$ contains the cup and the prompts $\{y_{orig}, y\}$ guide the optimization toward transformation of the existing object. The resulting statue heads $g(\theta)$ closely resemble their respective original cups appearance $g(\theta_{orig})$, indicating that DDS indeed inherits object lighting characteristics. For 2D object insertion, this behavior leads to unrealistic results: when DDS optimization is applied to an image with a copy-pasted cup, the final appearance remains inconsistent with the scene lighting due to this inheritance effect, as shown in App. L.

However, if we run DDS with $g(\theta_{orig})$ as the table image without the copy-pasted cup (second column, Fig. 4) with reference prompt $y_{orig} =$ «*a plate*», the initial optimized image $g(\theta)$ as the table with the copy-pasted cup (third column) with target prompt $y =$ «*a cup on a plate*», then the cup's appearance is improved (columns 4-6). In this setup DDS optimization becomes equivalent to «inpainting from scratch/generation of a cup on a plate», because the original image $g(\theta_{orig})$ does not contain a cup and prompts $\{y, y_{orig}\}$ lead the editing towards producing an object that is absent from $g(\theta_{orig})$. Since the initial optimized image $g(\theta)$ already contains a cup, DDS optimization is prevented from generating an arbitrary cup and is instead guided towards a result similar to the original. Moreover, DDS does not inherit the erroneous lighting of the copy-pasted cup, since the original image $g(\theta_{orig})$ is a realistic image of a table. Given that image inpainting using diffusion models improves objects appearances (Sec. 2.2), the final cup appearance is realistic.

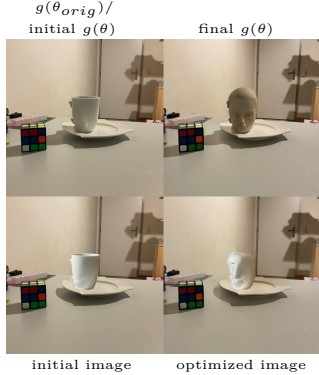

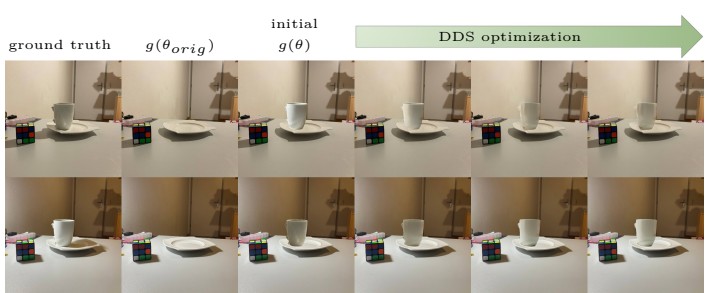

Figure 3: **DDS appearance inheritance.** The figure represents DDS editing transforming a «cup» into a «statue head». The left column shows the reference images $g(\theta_{\mathrm{orig}})$ with corresponding prompt $y_{\mathrm{orig}} =$ «*a cup on a plate*» (top: correct lighting; bottom: incorrect lighting). The optimized image is initialized as $g(\theta) = g(\theta_{\mathrm{orig}})$. The right column shows the final optimized image $g(\theta)$ after DDS optimization with target prompt $y =$ «*a statue head on a plate*».

Figure 4: **DDS refines object appearance after insertion.** Each row shows a separate DDS optimization experiment using the proposed initialization. The first column is ground truth The second column shows the reference image $g(\theta_{\mathrm{orig}})$ with prompt $y_{\mathrm{orig}} =$ «*a plate*». The third column shows the initialization of the optimized image $g(\theta)$, obtained by inserting a cup with incorrect lighting, with target prompt $y =$ «*a cup on a plate*». Subsequent columns show DDS optimization progressively correcting the cup's appearance using the lighting from $g(\theta_{\mathrm{orig}})$

## 3.4 DDS Optimization

In Sec. 2.2, $g(\theta)$ was described as an image. In practice, modern diffusion models operate in latent space to improve computational efficiency while preserving generation quality. Accordingly, $g(\theta)$ denotes a latent representation obtained by encoding an image with a Variational Autoencoder (VAE) (Kingma et al., 2013). SDS optimization is therefore performed in latent space, where the latent vector is optimized using the SDS loss and then decoded into image space. Direct optimization in image space often introduces noise and

artifacts due to inherent ambiguities (Tang et al., 2023). Notably, the same limitation appears in the context of 3DGS editing using SDS/DDS objectives (Xiao et al., 2024; Chen et al., 2024).

We propose a two-stage optimization strategy to resolve this problem. For clarity, we describe the idea in the context of SDS image generation, though it naturally extends to DDS image editing described later. The method is inspired by the observation that SDS optimization in latent space does not produce noisy images (Poole et al., 2022). For 2D generation using SDS, we proceed as follows: first, initialize the 2D image $x$ with random Gaussian noise, encode this noisy image $x$ into latent $z$, and optimize only the latent $z$ (keeping the image $x$ fixed) for several iterations («step 1»). Then, decode the optimized latent $z'$ (which is no longer noisy) from «step 1» back into image space into $x'$, and apply several steps of MSE optimization between the image $x$ and the $x'$ («step 2») to make the image $x$ closer to the noise-free $x'$. Finally, we repeat these two steps for several iterations. This approach significantly reduces artifacts and improves efficiency, as it avoids backpropagation through the VAE. For DDS optimization, the only difference lies in «step 1» of the algorithm, where the DDS objective replaces the SDS objective, while «step 2» remains the same. Experimental results demonstrating our method's effectiveness for 3DGS are shown in Sec. 4.5. The Python-like pseudocode 5 shows the 2-step-SDS for image generation. The 2-step-DDS 3DGS editing pseudocode and extra 2D image examples are provided in App. G.

### 3.5 Scene Shadows

After object insertion, we assume that scene Gaussians can only become darker due to the introduction of new shadows. To model such effects, we introduce a set of optimizable parameters $\alpha_1, \ldots, \alpha_n$, each associated with a scene Gaussian. Each parameter $\alpha_i$ modifies the color $c_i$ of the corresponding Gaussian as $c_i' = c_i - \text{Clip}(\alpha_i, [0, 1])$, where $\alpha_i$ is clipped to the interval $[0, 1]$ to ensure that colors can only darken. Since objects generally affect only nearby Gaussians, we empirically define the affected neighborhood as those within $N = 3$ bounding boxes around the object. Furthermore, only a limited subset of Gaussians within this region should actually darken. To enforce this, we retain only the top $\alpha = 30\%$ darkest shadow parameters every 500 optimization steps.

```python
image = random_image()
image_optimizer = get_optimizer(image, image_lr)
diffusion_model = diffusion.load_model()
prompt = get_prompt()
for nstep in num_iterations // (num_steps_latent +
    num_steps_image):
  latent = diffusion_model.autoencoder.encode(image.detach())
  for latent_step in num_steps_latent: # <<step 1>>
    # estimate SDS gradient
    g_latent = sds_grad(latent, prompt, <other arguments>...)
    # Update latent with SGD
    latent -= latent_lr * g_latent
  decoded_optimized_latent =
      diffusion_model.autoencoder.decode(latent)
  for image_step in num_steps_image: # <<step 2>>
    g = mse_grad(image, decoded_optimized_latent)
    image_optimizer.step(g)
return image
```

Figure 5: **Python-like pseudocode of 2-step-SDS.** Given a text `prompt`, the image is optimized by alternating latent-space SDS updates and pixel-space refinement steps. The `sds_grad(·)` samples $t, \varepsilon$ and computes the SDS gradient (Eq. 2).

## 4 Experimental results

### 4.1 Datasets

To the best of our knowledge, there are no publicly available datasets for evaluating 3DGS object insertion and relighting; therefore, we collected a dataset comprising both realistic and synthetic scenes for evaluation, described in detail below.

**Synthetic Data.** The dataset is built upon SceneNet (Handa et al., 2015), with additional objects sourced from BlenderKit (BlenderKit Online Community, 2024) and textures from Freepik (Freepik, 2024). The dataset includes 10 indoor scenes: *bathroom_1, bathroom_2, bedroom_1, bedroom_2, kitchen_1, kitchen_2, living_room_1, living_room_2, office_1, office_2*. Each scene contains a single placed object, such as "caution wet floor sign" in *bathroom_1*, "laundry basket" in *bathroom_2*, etc. We consider three rendering settings for evaluation: *object, scene*, and *object + scene* (used as ground truth for relighting), resulting in 30

image sets renders — 3 per scene across 10 indoor scenes (object only, scene only, and object within scene). For further details on the synthetic data and specific rendering settings refer to the App. E.

**Real Data.** We use the Specular AI (SpecularAI, 2025) mobile app to capture real scenes. For each object, we record two trajectories: one along a large circle and another along a small circle around the object. For each scene, we follow the same strategy with capturing along small and large circles, but the circle centers are in the location where the object was placed. In total, we collect three scenes and three objects, each capture consisting of approximately 300 images.

### 4.2 Implementation details

Our experiments are conducted on a single NVIDIA V100 32GB GPU. We use DN-Splatter (Turkulainen et al., 2024) to train 3DGS for synthetic data and splatfacto (Tancik et al., 2023) for real data. Our implementation is based on the nerfstudio (Tancik et al., 2023) framework. The prompts describing scenes ($y_{orig}$ in our notation, Eq. 3) and objects inside scenes ($y$ in our notation, Eq. 3) are user-defined.

For diffusion personalizations, we use Stable Diffusion 2.1 (Rombach et al., 2022) from Hugging Face (Wolf, 2019). We first render 32 images of the object 3DGS from random positions and process them using IC-Light (Zhang et al., 2025), following the procedure described in Sec. 3.2. For class preservation (Sec. 3.2), we generate 32 images of the same object class. For rough diffusion personalization, we randomly select an IC-Light-generated image with a probability of 0.7 and a class-preservation image with a probability of 0.3 during optimization. The training time of this stage is around 14 minutes per object. We fine-tune LoRA (Hu et al., 2022) with rank 4 for 1000 iterations and a batch size of 4 images, using the AdamW optimizer with a weight decay of $10^{-2}$, a learning rate of $10^{-4}$, and a constant learning rate scheduler. For texture-preserving diffusion personalization, we use the same images and training parameters. Examples of texture preservation are shown in Fig. 6. This stage reqires approximately 11 minutes per object.

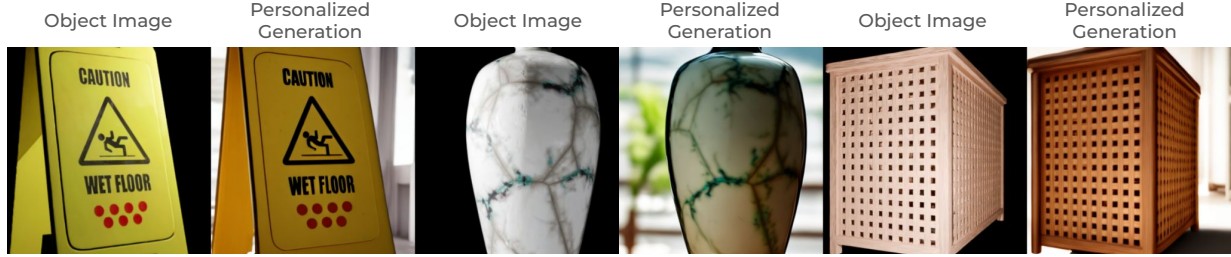

Figure 6: **Diffusion personalization for texture preservation.** Generation example of three different texture-preserving personalization models. The necessary object patters, such as text, decorative patterns are preserved.

For the second part, we initialize the object colors as the mean of all object colors and set its spherical harmonics (SH) coefficients to zero. Since the object is already represented in 3DGS, directly transferring its gaussian' parameters ensures correct geometry insertion, requiring adjustments only for colors and spherical harmonics (SH). The algorithm performs $3,000$ training steps ($1,000$ for 2-step-DDS and $2,000$ for refinement) and progressively includes object SH coefficients (discussed in Sec. 2.1) related to a higher SH degree in optimization every *sh_degree_interval* = 500 steps. We set *steps_image* = 64, *steps_latent* = 16, and *guidance_scale* = 10.0. We use the rough diffusion personalization model to improve object appearance. During the refinement phase, we use the texture-preserving diffusion model to maintain object identity and further enhance realism. The *steps_image* = 512 and we linearly decrease SDEdit timesteps from $[0.50, 0.02]$. To further enhance diffusion model geometry understanding, we use Depth ControlNet from Hugging Face durint both phases and set *controlnet_conditioning_scale* to 1.0. During the *step 1* of 2-step-DDS (latent optimization), we set *latent_lr* = 0.1. Object colors are optimized using Adam with a learning rate of 0.0025 and exponential learning rate decay, while object spherical harmonics are optimized with Adam at a learning rate of 0.0025/32. For image loss, we use the default nerfstudio (Tancik et al., 2023) optimization parameters of L1-Loss and SSIM. We simultaneously render $N = 8$ images and process them with diffusion

| Dataset | Metric | D3DR | Copy-Paste | LBM | TIP-Editor | R3DG | Instruct-GS2GS |
|---------|--------|------|-----------|-----|-----------|------|---------------|
| synthetic | $PSNR_{part}$ (↑) | **11.966** | 6.519 | 10.075 | 6.960 | 8.598 | 6.892 |
| | $PSNR_{cropped}$ (↑) | **18.039** | 13.032 | 16.271 | 12.502 | 14.454 | 13.360 |
| | $SSIM_{cropped}$ (↑) | **0.640** | 0.582 | 0.638 | 0.439 | 0.449 | 0.526 |
| | CTIS (↑) | **0.646** | 0.642 | 0.643 | 0.619 | 0.639 | 0.644 |
| | DTIS (↑) | **0.529** | **0.529** | 0.526 | 0.507 | 0.527 | **0.529** |
| real | CTIS (↑) | **0.643** | 0.638 | 0.638 | 0.625 | 0.623 | 0.641 |
| | DTIS (↑) | **0.510** | 0.505 | 0.506 | 0.497 | 0.501 | **0.510** |

Table 1: **Comparison with other methods.** *"Dataset"* column indicates the dataset used for evaluation, and *"Metric"* column lists the evaluated metrics. Other columns correspond to different methods, with values averaged across all scenes within each dataset. **Bold** numbers indicate the best performance. (↑) denotes that higher values are better, while (↓) indicates that lower values are better. D3DR achieves the best results on $PSNR_{part}$ and $PSNR_{cropped}$ for the synthetic dataset, on CTIS and DTIS for the real dataset.

| Metric | D3DR | Copy-Paste | LBM | TIP-Editor | R3DG | Instruct-GS2GS |
|--------|------|-----------|-----|-----------|------|---------------|
| Training time, minutes | 40 | **0** | 24 | 140 | 185 | 37 |
| Storage, GB | **0.076** | **0.076** | **0.076** | 0.097 | 0.955 | **0.076** |
| $N_{gaussians}$, $10^6$ | **0.330** | **0.330** | **0.330** | 1.870 | 1.970 | **0.330** |

Table 2: **Resources comparison.** *"Metric"* lists the evaluated metrics. Other columns show different methods, whose values are averaged across scenes. **Bold** indicates the best performance.

models, using 2-step-DDS or refinement phases, and then optimize 3DGS parameters. In total, the second part of the framework takes around 15 minutes per scene.

## 4.3   3DGS Object Insertion Evaluation

We compare our approach against naive copy-pasting, TIP-Editor (Zhuang et al., 2024), Relightable 3D Gaussian (R3DG) (Gao et al., 2025), Instruct-GS2GS (Vachha & Haque) and Latent Bridge Matching (LBM) (Chadebec et al., 2025). Baseline configurations are provided in the App. A.

The quantitative comparison is presented in Tab. 1. We report $PSNR_{part}$, which measures PSNR over object pixels only, as well as $PSNR_{cropped}$ and $SSIM_{cropped}$, which measure PSNR and SSIM within the object bounding box. Additionally, following previous works (Shahbazi et al., 2024; Zhong et al., 2024), we compute CLIP-based metrics. *CTIS* denotes the average cosine similarity between the CLIP features of the target prompt and the target images (i.e., rendered scenes with the inserted, processed objects). *DTIS* measures the average cosine similarity between the difference in CLIP features of the target (rendered scene with the inserted, processed object) and initial (rendered scene) images, and the difference in CLIP features of their corresponding prompts, reflecting the alignment of transformations. Both *CTIS* and *DTIS* are normalized to the range $[0, 1]$. For real datasets, we do not provide PSNR and SSIM metrics due to inaccuracies in the ground-truth poses. Example results are shown in Fig. 7, and additional ones are provided in App. J. Tab. 1 shows that our method consistently outperforms TIP-Editor, R3DG, LBM, and Copy-Paste across metrics measuring object harmonization $PSNR_{part}$, $PSNR_{cropped}$, $SSIM_{cropped}$, and insertion realism CTIS, and DTIS. It also outperforms InstructGS2GS on all metrics except DTIS, where both methods achieve the same score on the synthetic and real datasets. We report shadow-focused metrics in the Tab. 3 of App. B.

The Tab. 2 represents metrics regarding resources and training time. It indicates that our method trains nearly three times faster than TIP-Editor and R3DG, while requiring less storage and fewer Gaussians to represent the scenes. The Instruct-GS2GS requires 7.5% less training time than our method, however the latter has better overall evaluation performance and preserves object identity more effectively. While LBM

requires about 40% less training time than our method, it lacks multi-view consistency and does not generate scene-aware shadows, both of which are essential for realistic 3D object insertion.

TIP-Editor struggles with generating large objects because it reconstructs the object 3DGS from scratch using SDS, whereas our method directly leverages existing 3DGS objects representations. R3DG, on the other hand, suffers from issues related to incident light and albedo, as discussed in Sec. 2.3, leading to suboptimal object insertion quality. LBM does not incorporate shadow reconstruction into its predictions. Moreover, its results lack multi-view consistency and contain artifacts, such as the black line along the edge of the *Caution Wet Floor* sign shown in Fig. 7. InstructGS2GS fails to preserve essential object details, such as the text on the *Caution Wet Floor* sign, and produces unrealistic relighting, likely due to the known limitations of the underlying InstructPix2Pix model (Bashkirova et al., 2023).

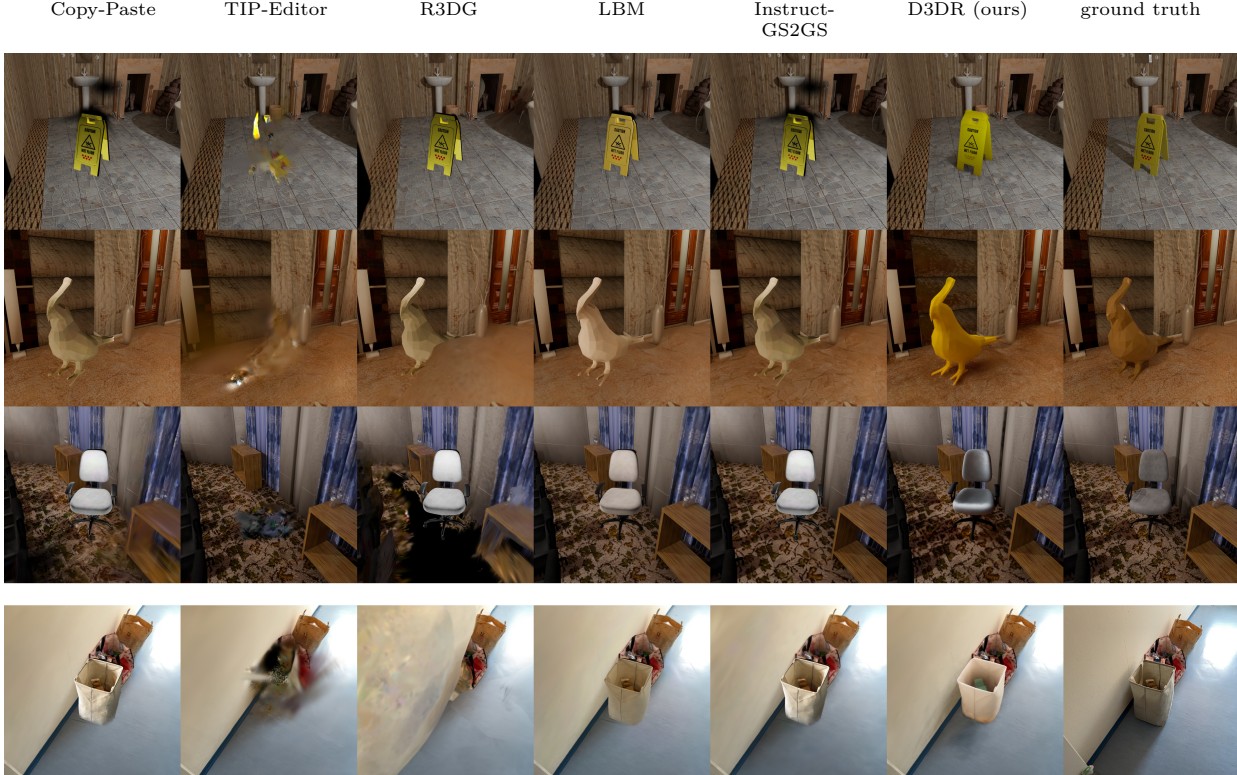

Figure 7: **Comparison with other methods.** Rows show different scenes (top to bottom: bathroom_1, living_room_1, office_1, fabric bin in corridor), and columns show different methods. Our method improves object appearance. A zoomed-in view is provided in Fig. 20.

## 4.4 Object Insertion with Different Diffusion Models

To demonstrate the effectiveness of training-free diffusion models for the 3DGS object insertion problem, we run our pipeline on *bathroom_1* using different diffusion models. Due to limited computational resources, we present results only for Stable Diffusion (Rombach et al., 2022) versions 1.5, 2.0, and 2.1. As shown in Fig. 8, all three models effectively adjust object lighting after insertion, producing realistic results.

To verify that this observation also holds for versions 3.0 and 3.5, whose diffusion processes are formulated as rectified flow (Liu et al., 2022), we run DDS optimization on our 2D toy cup dataset, as depicted in Fig. 9. We report results for Stable Diffusion 1.5, 2.0, and 2.1 (Rombach et al., 2022), as well as for versions 3.0 and 3.5 (Esser et al., 2024). Further details on applying DDS optimization to diffusion models formulated as rectified flow are provided in App. H. As illustrated in Fig. 9, DDS optimization consistently improves the

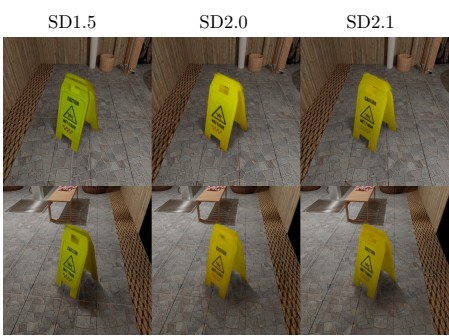

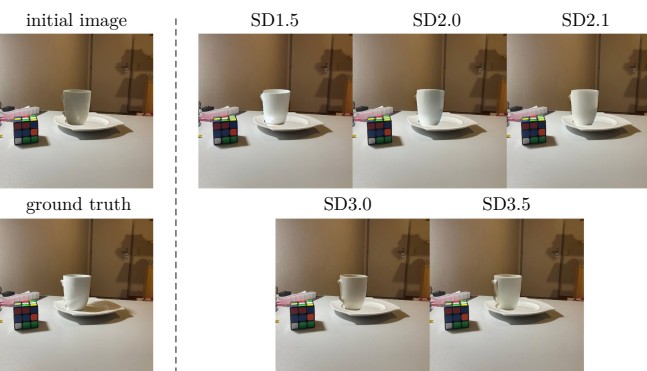

Figure 8: **3D object insertion with different diffusion models.** Each column corresponds to a different diffusion model, and each row shows a different viewpoint. All models produce realistic lighting and shadows, though SD1.5 introduces a slight green tint and SD2.0 fails to reproduce the *"Caution Wet Floor"* text on the sign.

Figure 9: **Object realism improvement across diffusion models.** The left column shows the initial image with the copy-pasted cup and the ground truth. The right panels display DDS optimization results for different Stable Diffusion versions (1.5, 2.0, 2.1, 3.0, and 3.5), demonstrating consistent enhancement of object lighting and realism.

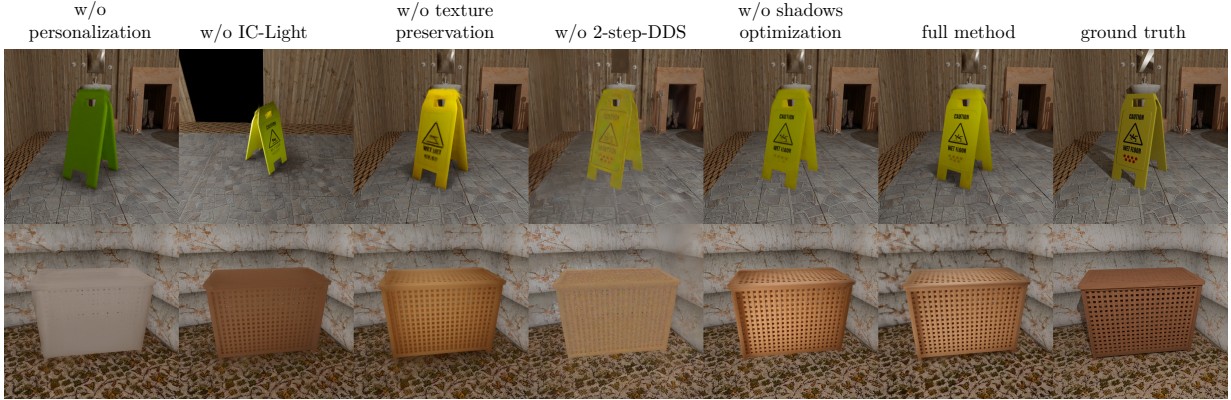

Figure 10: **Ablation studies.** Different columns represent different ablation experiments on *bathroom_1* scene. For *no iclight* setting a different view is shown to highlight unrealistic object appearance.

cup's appearance during insertion across all models. This demonstrates that the inherent ability of diffusion models — enhancing object lighting after insertion — persists across different variants and formulations.

## 4.5 Ablation studies

To analyze the contribution of each component of our method, we perform an ablation study on the *bathroom_1* scene by selectively disabling individual parts of our pipeline. We analyze the effects of diffusion personalization, IC-Light images, texture preservation, and the proposed 2-step DDS optimization. As shown in Fig. 10, each component is essential for achieving realistic object appearance and consistent lighting.

The ablation results in Fig. 10 highlight the contribution of each component of our pipeline. **(a) Diffusion personalization:** disabling this module causes the inserted object to deviate substantially from the original, losing essential texture details and even its characteristic color (e.g., the yellow hue of the *Caution Wet Floor* sign). **(b) IC-Light:** removing IC-Light images during personalization leads to overfitting to a single illumination setup, resulting in unrealistic appearance that closely resembles the initial, unadapted object. **(c) Texture preservation:** omitting our texture-preserving diffusion model causes a loss of fine-grained surface details, such as the printed lettering on the sign. **(d) 2-step DDS:** replacing our method with standard DDS optimization introduces visible noise artifacts. Our approach eliminates the need for gradient

computation through the VAE, yielding a substantial efficiency gain — 16,000 DDS optimization steps complete in about 7 minutes, whereas standard DDS requires roughly 40 minutes for 10,000 steps. **(e) Shadows:** Disabling the shadow parameters during optimization produces visual results that lack essential floor shadows. We calculate metrics in Tab. 4 of App. C

## 5 Conclusion

We have introduced D3DR, a diffusion-driven method for natural and consistent 3D object insertion within 3D Gaussian Splatting (3DGS) scenes. By optimizing object and scene parameters through a diffusion-based Delta Denoising Score (DDS) objective, our framework exploits the inherent priors of diffusion models to harmonize illumination and appearance without requiring any task-specific dataset. This reveals, for the first time, that diffusion models possess an intrinsic zero-shot capability to perform lighting-consistent object insertion and relighting in 3DGS representations. The proposed 2-step-DDS optimization reduces noise and improves stability and efficiency, while our personalized diffusion models — tailored for both relighting and texture preservation — enable accurate reproduction of object appearance and fine-grained surface details. Comprehensive evaluations across synthetic and real-world datasets demonstrate that D3DR substantially outperforms existing approaches in lighting realism, training efficiency, and visual coherence. We believe that this work uncovers a new zero-shot capability of diffusion models as 3D harmonizers, paving the way toward more realistic and versatile scene editing and object manipulation in neural graphics.

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

## A   Baselines Configuration

**TIP-Editor.**   We observe that TIP-Editor fails to generate objects when a scene 3DGS is trained with DN-Splatter, requiring us to use its original training pipeline instead. We follow their default training parameters, except for the object generation parameters, where we set $start\_gamma = 0.1$ and $end\_gamma = 0.5$, as the original settings fail to generate large objects. We also ensure that the scene description prompts remain identical between TIP-Editor and our method. The average total training time for TIP-Editor, excluding 3DGS scene training, consists of 20 minutes for scene personalization over 1,000 steps, 3 minutes for object personalization over 500 steps, 115 minutes for SDS generation over 5,000 steps, and 2 minutes for SDS refinement over 5,000 steps, totaling 140 minutes.

**R3DG.**   R3DG requires surface normals of 3D Gaussians, which are not available in our object 3DGS. To ensure a fair comparison, we adopt the R3DG training pipeline for both object and scene 3DGS. For scene training, we use the script provided for the Tanks and Temples dataset (Knapitsch et al., 2017), as it most closely resembles our dataset. We observe that the 3DGS model begins producing floaters after approximately 10,000 iterations; therefore, we save a checkpoint at this stage. We then continue with the second training phase for an additional 20,000 iterations using the official script. For object training, we employ the script provided for the NeRF Synthetic dataset. The total average training time for R3DG — excluding 3DGS scene and object training — includes 150 minutes for scene physical parameter learning and 35 minutes for object physical parameter learning, totaling 185 minutes per scene.

**Instruct-GS2GS.**   To adapt Instruct-GS2GS (Vachha & Haque) to our setting, an object is first inserted into the scene, and then the combined 3DGS is rendered from the evaluation cameras positions followed by the original training pipeline. Following the authors' recommendation, we evaluate 15 candidate prompts ( *"make the object appearance realistic", "make the object appearance match the surroundings", "make the object appearance match the surrounding lighting", "make the appearance realistic", "make the lighting in the picture realistic", "make object insertion look realistic", "make the object appearance realistic and consistent", "improve object appearance and add missing shadows", "the object was copy-pasted, improve its appearance and add missing shadows", "make the appearance of caution wet floor sign realistic", "make the lighting and shadows realistic", "make the object lighting and shadows realistic", "make the lighting consistent and realistic, as if the object was originally part of the scene", "match the object lighting and shadows to the surrounding environment." "make the appearance of the object to look realistic and consistent with the scene", "adjust the lighting so the inserted object matches the lighting of the scene",* ) using InstructPix2Pix and select the instruction *"make the object appearance correct"* for 3DGS editing. The training process makes 7,500 iterations which takes approximately 37 minutes.

**Latent Bridge Matching.**   Latent Bridge Matching (LBM) is originally developed for 2D object harmonization. To adapt it to our setting, we insert the 3DGS object into the scene, render the combination from the training views, and apply LBM to the rendered images (approximately 17 minutes per scene) to construct a training dataset. We then optimize the colors and spherical harmonics (SH) parameters of both the scene and the object for 10,000 iterations, following the DN-Splatter pipeline. This optimization requires about 7 minutes per scene. In total, the method requires approximately 24 minutes per scene.

## B   Shadow Evaluation

We report $\text{PSNR}_{shadows}$ and $\text{SSIM}_{shadows}$, defined as PSNR and SSIM on pixels near the object (within $1.20\times$ its bounding box) with object pixels excluded, and $\text{PSNR}_{background}$ and $\text{SSIM}_{background}$, defined as PSNR and SSIM on background pixels.

| Metric | D3DR | Copy-Paste | LBM | TIP-Editor | R3DG | Instruct-GS2GS |
|---|---|---|---|---|---|---|
| $\text{PSNR}_{shadow}$ ($\uparrow$) | **19.993** | 19.023 | 19.544 | 11.899 | 14.899 | 17.923 |
| $\text{PSNR}_{background}$ ($\uparrow$) | 23.383 | **24.169** | 23.533 | 16.786 | 15.232 | 20.238 |
| $\text{SSIM}_{shadow}$ ($\uparrow$) | 0.890 | **0.924** | 0.894 | 0.768 | 0.738 | 0.802 |
| $\text{SSIM}_{background}$ ($\uparrow$) | 0.910 | **0.948** | 0.902 | 0.815 | 0.679 | 0.780 |

Table 3: **Shadows Metrics on Synthetic Dataset.** *"Metric"* column lists the evaluated metrics. Other columns correspond to different methods, with values averaged across all scenes of the synthetic dataset. **Bold** numbers indicate the best performance. ($\uparrow$) denotes that higher values are better. D3DR achieves the best results on $\text{PSNR}_{shadow}$ and competitive results on $\text{SSIM}_{shadow}$.

| Method | PSNR, part | PSNR, cropped | PSNR, shadows | PSNR, background | SSIM, shadows | SSIM, background | SSIM, cropped | CTIS | DTIS |
|---|---|---|---|---|---|---|---|---|---|
| D3DR | **11.966** | **18.039** | **19.993** | 23.383 | 0.890 | 0.910 | 0.640 | **0.646** | **0.529** |
| D3DR w/o shadow | 11.762 | 17.730 | 19.983 | **24.879** | **0.926** | **0.942** | **0.663** | 0.645 | 0.528 |

Table 4: **Shadow parameter comparison.** Comparison of the method with the shadow parameter from Sec. 3.5 enabled and disabled. Method specifications are listed in the first column, followed by the evaluation metrics. All metrics are computed on the synthetic dataset.

The Tab. 3 demonstrates the shadow evaluation results. The method achieves superior performance on $\text{PSNR}_{shadow}$ and competitive $\text{SSIM}_{shadow}$, indicating high shadow fidelity relative to ground truth. Copy-Paste and LBM achieve higher scores on background metrics because our diffusion-guided shadow optimization introduces small appearance changes in nearby scene regions. While these adjustments improve shadow realism, they can slightly reduce global similarity metrics.

## C  Ablation on Shadow Parameter

We perform training with shadow parameter introduced in Sec. 3.5 disabled and calculate the metrics in Tab. 4. Disabling the shadow parameters during optimization degrades performance across nearly all metrics measuring object harmonization quality. The higher background metric values of the shadow-disabled method occur because our full method darkens scene Gaussian colors near the object due to diffusion-guided optimization, while producing realistic shadows.

## D    Limitations

Although the method is robust and performs well on both synthetic and realistic scenes, it has several limitations, mainly inherited from diffusion models. **(1) Albedo change:** the method may alter the object albedo, primarily due to IC-Light generation. An example is shown in the top row of Fig. 11, where a *"suitcase"* is inserted into the scene. The IC-Light result (middle image) produces the object with colors brighter the original ones and personalized models remember those colors. This issue can be mitigated by using more precise prompts or generating multiple samples and discarding those with incorrect albedo. **(2) Complex textures:** the method struggles with complex, highly textured objects. For example, when inserting a "rock" (middle row of Fig. 11), the personalization model oversmooths the rough texture (middle image). Training a controller network, such as ControlNet (Zhang et al., 2023), to better preserve texture details—rather than relying solely on per-object personalization—could improve performance. **(3) Implicit lighting direction** under uniform illumination with ambiguous light direction, shadows generated by our method can appear faint. For example, when inserting a "cup" (bottom row of Fig. 11), the desk appearance is almost constant, causing 2-step-DDS to struggle with shadow generation (middle image). Personalizing the coarse diffusion model to both the scene and the object may help alleviate this issue. **(4) Dynamic scenes:** the current method is not designed for dynamic scenes. Incorporating video diffusion models (Wan et al., 2025) could address this limitation.

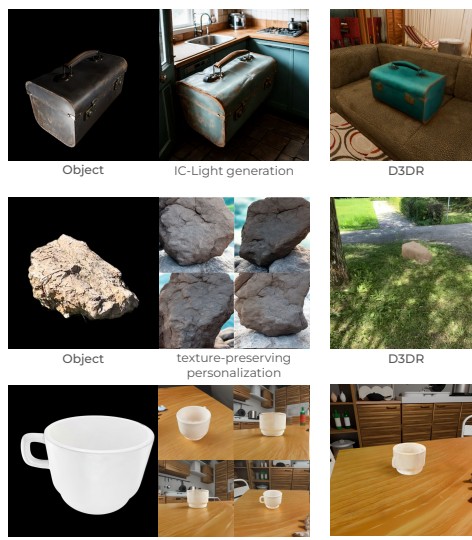

Figure 11: **Limitations.** Each row represents failure cases. The first column represent object to be inserted, the second column represent the cause of the problem, the third row represents the result of object insertion with D3DR. Zoom-in for better view.

## E    Rendering Details and Point Cloud Generation

In this section, we provide detailed descriptions of the rendering settings and the point cloud generation methods used for the datasets collection.

### E.1    Rendering Settings

We consider three rendering settings — when we render only object, object + scene together, and only scene:

- **Object Rendering**: The object is placed at the origin. A set of point lights is placed around the object, with the number of lights increasing for larger objects. The camera follows a circular trajectory with random variations in distance and look-at position to introduce natural perturbations. Both images and object masks are generated.

- **Object + Scene Rendering**: The object is placed in a defined place in the scene. Then camera moves around the object, with small random perturbations, such that the trajectory is similar to camera trajectory in *object rendering* setting. Both the object and the scene are rendered. Additionally, we render the object masks.

- **Scene Rendering**: The object is hidden, and the camera follows the same trajectory as in the *object + scene rendering* setting. Only the scene is rendered.

### E.2 Point Cloud Generation

Gaussian Splatting requires a sparse set of points for initialization. Existing methods, such as those in BlenderNeRF Raafat (2024), typically place points only at the mesh corners, which can result in an uneven distribution on the surface. We propose an improved point cloud generation approach of an arbitrary Blender scene, which can be either *object, scene or object + scene* in our notation. The method integrates three sampling strategies, described below.

1. **Surface Area Sampling**: A blender scene item (e.g. floor, wall, chair which belong to the scene) is sampled based on its surface area, using $Volume^{2/3}$ instead of ordinary area to avoid over-representing thin structures like plant leaves. Then, a triangle is selected from the item's mesh proportional to its area, and a point is sampled uniformly on the triangle.

2. **Uniform Triangle Sampling**: A scene item is uniformly sampled, followed by the uniform sampling of a triangle from its mesh. Finally, a point is sampled uniformly on the triangle.

3. **Bounding Box Sampling**: A point is sampled within the scene's bounding box, the closest mesh triangle is found, and a point is sampled uniformly on that triangle.

### E.3 Rendering and Point Cloud Generation Details

We use the CYCLES renderer with 256 samples per image, generating 250 images per setting. Object masks are rendered for *object* and *object + scene* settings.

For sparse point clouds, we sample:

- 10,000 points for *object*

- 5,000 points for *object + scene*

- 50,000 points for *scene*

Dataset images are shown in fig. 12.

The mapping between our scenes names and SceneNet Handa et al. (2015) names is (the first is ours, and the second is from SceneNet):

- *bathroom_1* $\longleftrightarrow$ bathroom_5,

- *bathroom_2* $\longleftrightarrow$ bathroom_28

- *bedroom_1* $\longleftrightarrow$ bedroom_3,

- *bedroom_2* $\longleftrightarrow$ bedroom_27

- *kitchen_1* $\longleftrightarrow$ kitchen_35,

- *kitchen_2* $\longleftrightarrow$ kitchen_76

- *living_room_1* $\longleftrightarrow$ living_room_11,

- *living_room_2* $\longleftrightarrow$ living_room_33

- *office_1* $\longleftrightarrow$ office_14,

- *office_2* $\longleftrightarrow$ office_23

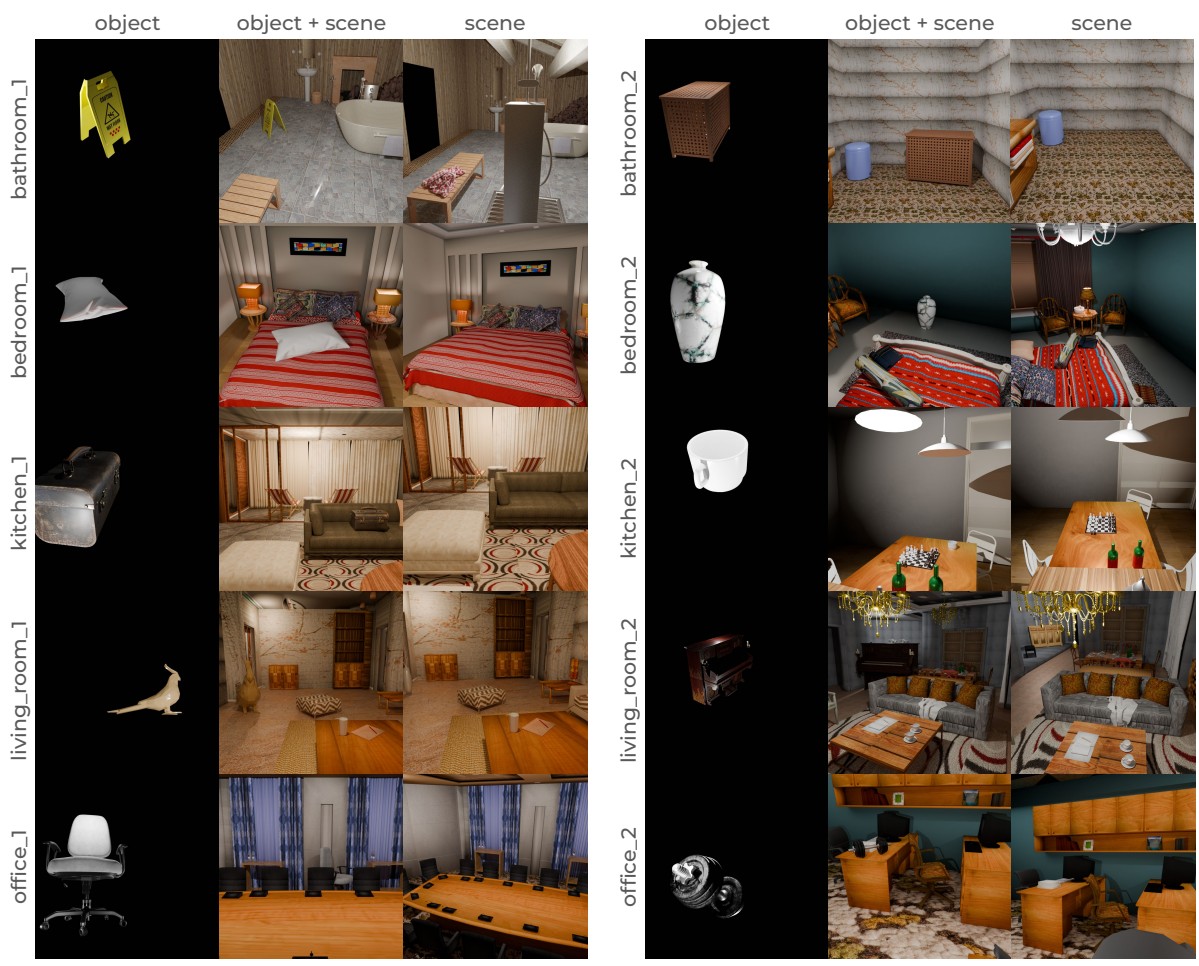

Figure 12: **Our dataset.** The dataset comprises 30 sets of images and point clouds, divided into three categories: 10 sets of objects, 10 sets of objects in scenes, and 10 sets of standalone scenes. Each item represents an image from the related renderings. Rows correspond to different images captured within the same scene, while columns group images by category: objects, objects in scenes, and solely scenes.

## F   Tiny Cup Dataset

We use a small dataset of four real images and two artificially created images depicted on Fig. 13 to analyze how DDS relies on object appearance for editing. The real images show a cup in a room under two different lighting conditions, with an additional pair of images showing the same room under the same lighting conditions but without the cup. The artificial images are created by copy-pasting the cup, captured under one lighting condition, into an image of the room with a different illumination.

## G   2-step-SDS

### G.1   Algorithm

Our detailed 2-step-DDS optimization procedure is provided as a pseudocode in algorithm 1. Figure 14 explains 2-step-SDS algorithm for noiseless image generation in pixel space.

### G.2   Image Generation

We conduct image generation in image space using our proposed 2-step-SDS approach, with results shown in Fig. 15. Classical SDS optimization in image space introduces noticeable noise artifacts, whereas both SDS optimization in latent space and our 2-step-SDS method effectively mitigate this issue. We set the classifier-free guidance coefficient to $\omega = 15$ and perform $1,000$ optimization steps for each method. SDS latent optimization and 2-step-SDS complete image generation in approximately 1 minute, while classical SDS in image space requires 2 minutes.

## H   Rectified Flow Delta Denoising Score

Rectified Flow (Liu et al., 2022) defines a transport path between two distributions, $Z_0$ and $Z_1$, which represent Gaussian noise and the image distribution respectively. Let $x_0$ be a Gaussian noise sample and $x_1$ an image. The intermediate sample at timestep $t \in [0, 1]$ is $x_t = tx_0 + (1 - t)x_1$. To solve the transport

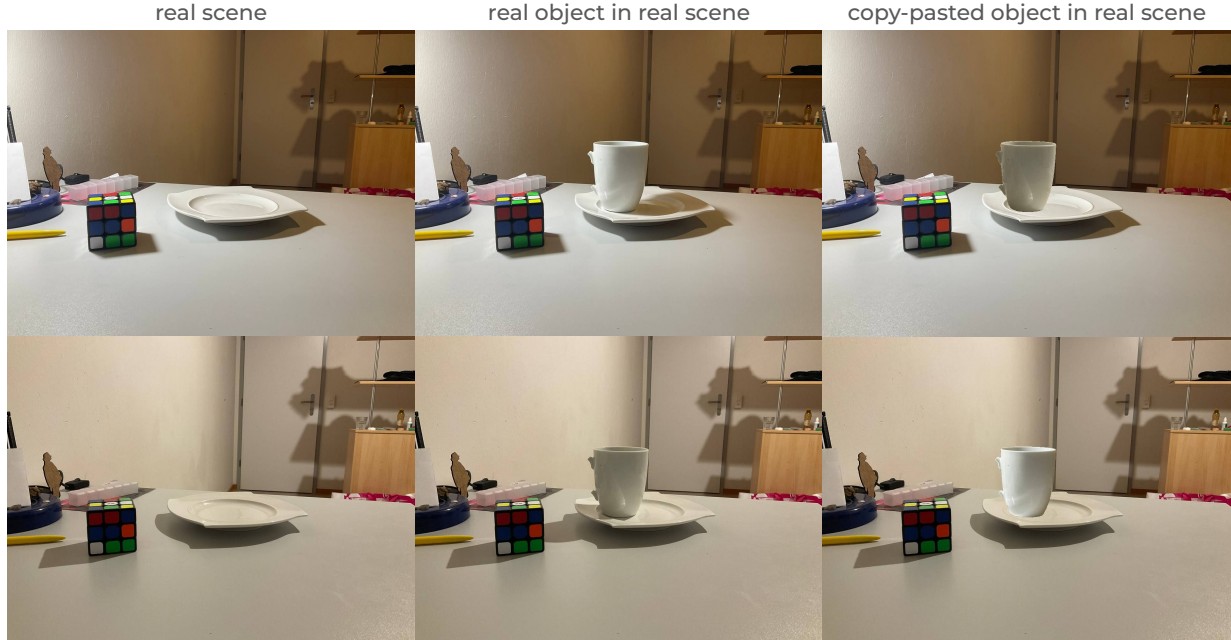

Figure 13: **Tiny cup dataset.** This dataset represent the object (cup) in the scene (room) under various illumination conditions.

---

**Algorithm 1** 2-step-DDS optimization

---

1: **procedure** 2-STEP-DDS(unet, vae, $y_{tgt}, y_{init}$,
    $\theta_{tgt}, \theta_{init}$,
    steps_latent, steps_image, num_iters, lr)
2:     $\theta = \theta_{tgt}$
3:     optimizer = Adam($\theta$.state_dict())
4:     $x_{init}$ = vae.encode($g(\theta_{init})$).detach()
5:     **for** $i$ $in$ $[1 \ldots \lceil \frac{num\_iters}{steps\_image} \rceil]$ **do**
6:         latent = vae.encode($g(\theta)$).detach()
7:         # step 1: SGD latent optimization
8:         **for** $j$ $in$ $[1 \ldots steps\_latent]$ **do**
9:             $t \sim \mathcal{U}(1, T)$, $\varepsilon \sim \mathcal{N}(0, I)$
10:            $z_{tgt} = \alpha_t \text{latent} + \sigma_t \varepsilon$
11:            $z_{init} = \alpha_t x_{init} + \sigma_t \varepsilon$
12:            calculate $\nabla L_{dds}$ using unet, $t, z_{tgt}, y_{tgt}, z_{init}, y_{init}$
13:            latent = latent $- lr \cdot \nabla L_{DDS}$
14:        **end for**
15:        # step 2: Adam $\theta$ optimization
16:        image_opt = vae.decode(latent)
17:        **for** $j$ $in$ $[1 \ldots steps\_image]$ **do**
18:            optimizer.zero_grad()
19:            loss = MSE($g_{tgt}(\theta, p)$, image_opt)
20:            loss.backward()
21:            optimizer.step() # $\theta$ is updated
22:        **end for**
23:    **end for**
24:    **return** $\theta$
25: **end procedure**

---

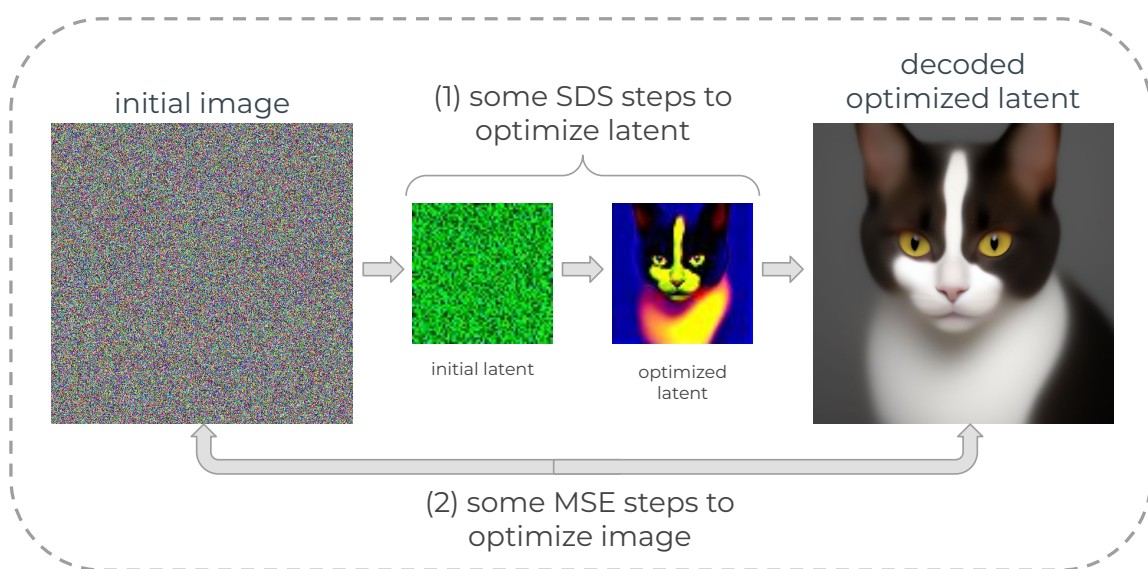

Figure 14: **Explanatory image of 2-step-SDS.** The algorithm begins by generating the `initial image` as random noise. It is then encoded into latent space to obtain the `initial latent`. Several optimization steps are performed on this latent (without modifying the initial image), resulting in the `optimized latent`. The `optimized latent` is then decoded back into image space (`decoded optimized latent`), and the `initial image` is updated to better match the `decoded optimized latent`. This is repeated for multiple iterations.

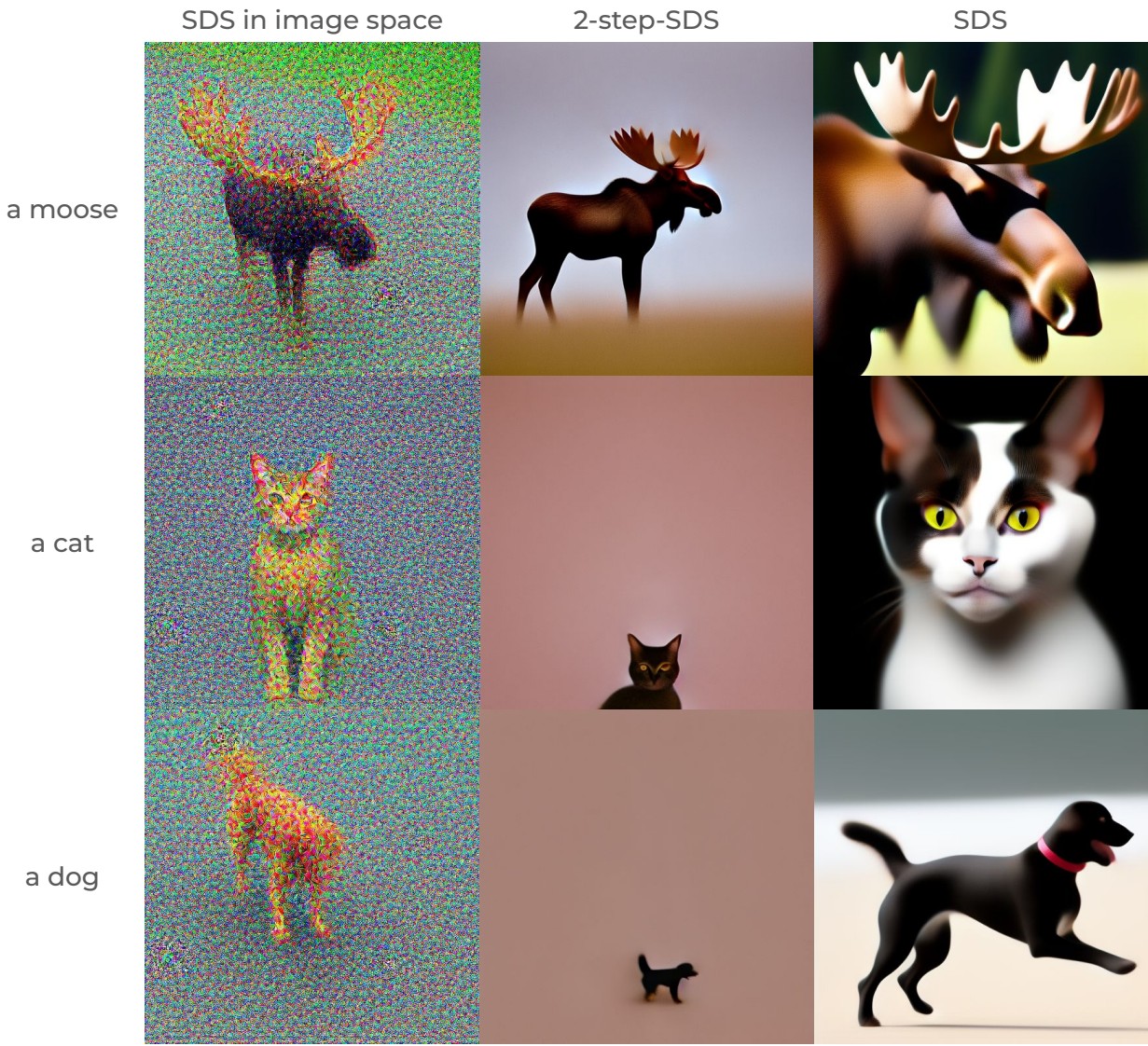

Figure 15: **2-step-SDS generation.** The first column shows results from classical SDS optimization in image space, the second column presents our 2-step-SDS optimization, and the third column illustrates classical SDS optimization in latent space. Each row corresponds to a different prompt.

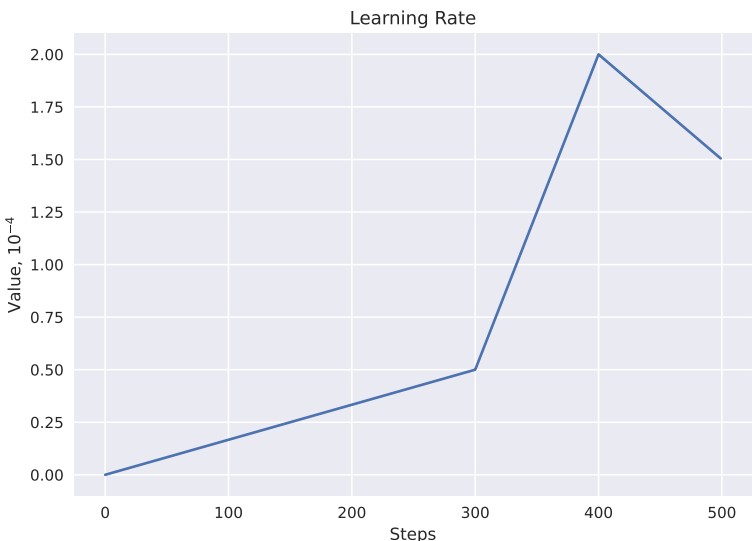

Figure 16: **Learning Rate for Rectified Flow DDS**. The figure illustrates the learning rate schedule used for DDS in Rectified Flow models. It increases linearly from 0.0 to $\frac{1}{4}max\_lr$ during the first 60% of optimization, then linearly increases from $\frac{1}{4}max\_lr$ to $max\_lr$ between 60% and 80%. Finally, it decreases linearly from $max\_lr$ to $\frac{3}{4}max\_lr$ over the last 20% of optimization.

problem from $x_0$ to $x_1$, the model learns the *velocity* field $v_\phi(x_t, t) = x_1 - x_0$. The training loss is defined as:

$$\int_0^1 \mathbb{E}\big[|v_\phi(x_t, t) - (x_1 - x_0)|^2\big], dt \tag{4}$$

The SDS loss is derived in the same way as in (Poole et al., 2022), which is done in (Yang et al., 2024):

$$\nabla L_{sds\_rf} = \mathbb{E}[v_\phi(x_t, t) - (x_1 - x_0)] \tag{5}$$

where constant terms are omitted. In our notation from Sec. 2, this becomes:

$$\nabla_\theta L_{sds\_rf} = \mathbb{E}\big[\varepsilon_\phi^\omega(t\varepsilon + (1-t)g(\theta), t) - (g(\theta) - \varepsilon)\big] \tag{6}$$

The DDS derivation follows the original formulation (Hertz et al., 2023) and is described in (Beaudouin et al., 2025). In our notation from Sec. 2, it is written as:

$$\nabla_\theta L_{dds\_rf} = \mathbb{E}\big[\varepsilon_\phi^\omega(t\varepsilon + (1-t)g(\theta), t) - \varepsilon_\phi^\omega(t\varepsilon + (1-t)g(\theta_{orig}), t) - (g(\theta) - g(\theta_{orig}))\big] \tag{7}$$

Directly optimizing this objective gives poor results. Following the learning rate scheduling strategy from (Beaudouin et al., 2025), we use the derived DDS with the learning rate schedule shown in Fig. 16. During training, the noise ratio linearly decreases from 0.95 to 0.05. The motivation behind this schedule is that initial noisy iterations do not provide accurate gradients, since the diffusion model cannot fully observe the object under such noise. In contrast, later iterations have much lower noise levels and therefore produce more reliable gradients.

## I  Per scene comparison

Tab. 5 and Tab. 6 present a comparison between methods for the synthetic dataset. Our approach achieves the best performance in nearly every scene in terms of $\text{PSNR}_{\text{part}}$, $\text{PSNR}_{\text{cropped}}$ and $\text{SSIM}_{\text{cropped}}$. It outperforms

| Metric | Method | bathroom_1 | bathroom_2 | bedroom_1 | bedroom_2 | kitchen_1 | kitchen_2 |
|---|---|---|---|---|---|---|---|
| PSNR$_{part}$ (↑) | D3DR | **11.027** | **10.591** | **14.102** | **9.608** | **13.396** | 5.549 |
| | LBM | 10.983 | 10.016 | 13.874 | 8.597 | 11.266 | **8.263** |
| | TIP-Editor | 9.801 | 7.586 | 6.709 | 3.724 | 7.382 | 2.530 |
| | R3DG | 9.062 | 9.742 | 8.670 | 7.480 | 9.302 | 5.735 |
| | Copy-Paste | 5.426 | 6.347 | 7.751 | 6.429 | 6.544 | 2.638 |
| | Instruct-GS2GS | 5.524 | 6.948 | 7.841 | 7.394 | 6.786 | 3.544 |
| PSNR$_{cropped}$ (↑) | D3DR | 17.306 | **16.100** | **19.528** | **15.472** | **19.020** | 10.853 |
| | LBM | **17.369** | 15.220 | 19.119 | 14.502 | 16.861 | **13.298** |
| | TIP-Editor | 14.592 | 12.876 | 12.215 | 8.530 | 12.635 | 6.699 |
| | R3DG | 14.900 | 15.383 | 13.548 | 13.211 | 14.866 | 10.998 |
| | Copy-Paste | 12.108 | 11.994 | 13.894 | 12.343 | 12.519 | 8.013 |
| | Instruct-GS2GS | 12.186 | 12.527 | 13.898 | 13.298 | 12.709 | 8.862 |
| SSIM$_{cropped}$ (↑) | D3DR | 0.550 | **0.546** | **0.791** | **0.693** | **0.620** | 0.640 |
| | LBM | **0.608** | 0.509 | 0.761 | 0.689 | 0.596 | **0.721** |
| | TIP-Editor | 0.416 | 0.374 | 0.456 | 0.346 | 0.394 | 0.557 |
| | R3DG | 0.451 | 0.432 | 0.586 | 0.510 | 0.273 | 0.488 |
| | Copy-Paste | 0.520 | 0.438 | 0.670 | 0.680 | 0.520 | 0.602 |
| | Instruct-GS2GS | 0.449 | 0.377 | 0.603 | 0.647 | 0.405 | 0.598 |
| PSNR$_{shadow}$ (↑) | D3DR | 18.949 | **16.536** | **18.777** | **26.066** | **21.076** | **14.404** |
| | LBM | **19.177** | 15.799 | 17.924 | 25.533 | 19.252 | 14.175 |
| | TIP-Editor | 14.710 | 12.673 | 11.692 | 5.525 | 10.773 | 1.621 |
| | R3DG | 13.674 | 15.496 | 12.364 | 19.388 | 14.934 | 13.572 |
| | Copy-Paste | 15.451 | 16.224 | 17.775 | 24.714 | 19.449 | 12.866 |
| | Instruct-GS2GS | 15.123 | 14.897 | 16.070 | 24.192 | 16.782 | 12.378 |
| PSNR$_{background}$ (↑) | D3DR | **24.185** | 17.056 | 24.671 | 27.880 | **24.035** | **23.101** |
| | LBM | 23.344 | 17.119 | 23.486 | 29.244 | 22.413 | 22.462 |
| | TIP-Editor | 20.262 | 13.917 | 18.153 | 12.946 | 14.240 | 12.399 |
| | R3DG | 13.574 | 15.387 | 12.802 | 19.203 | 15.441 | 14.987 |
| | Copy-Paste | 20.622 | **17.957** | **25.561** | **30.829** | 23.886 | 22.456 |
| | Instruct-GS2GS | 19.072 | 15.788 | 18.938 | 23.550 | 18.413 | 17.880 |
| CTIS (↑) | D3DR | 0.659 | 0.633 | **0.649** | **0.649** | **0.664** | 0.650 |
| | LBM | 0.657 | 0.635 | 0.646 | **0.649** | 0.659 | 0.648 |
| | TIP-Editor | 0.606 | 0.600 | 0.640 | 0.618 | 0.637 | 0.644 |
| | R3DG | 0.650 | 0.635 | 0.637 | 0.629 | 0.662 | **0.654** |
| | Copy-Paste | 0.659 | 0.634 | 0.648 | 0.630 | 0.662 | 0.649 |
| | Instruct-GS2GS | **0.661** | **0.636** | 0.648 | 0.647 | 0.660 | 0.649 |
| DTIS (↑) | D3DR | 0.559 | 0.531 | **0.505** | **0.535** | 0.528 | 0.517 |
| | LBM | 0.555 | 0.531 | 0.503 | 0.534 | 0.524 | 0.513 |
| | TIP-Editor | 0.515 | 0.504 | 0.502 | 0.504 | 0.502 | 0.514 |
| | R3DG | 0.556 | 0.535 | 0.504 | 0.515 | 0.530 | **0.521** |
| | Copy-Paste | 0.564 | 0.535 | 0.503 | 0.517 | **0.532** | 0.516 |
| | Instruct-GS2GS | **0.565** | **0.538** | **0.505** | 0.531 | 0.530 | 0.514 |

Table 5: **Synthetic Dataset full comparison 1.** The table is split into two parts due to its size; this is Part 1. The first column represents metric names, the second column represents method names, columns 3-8 represent per scene results across different methods. **Bold** numbers represent the best across methods for a scene. ↑ represents that the metric is better if the value is greater. D3DR outperforms other methods on PSNR$_{part}$, PSNR$_{cropped}$, SSIM$_{cropped}$, CTIS$_{cropped}$ almost on every scene.

the baselines on average in the CTIS, and performs the same on DTIS metric. Table 7 presents a comparison of methods on real scenes. Our approach outperforms the baselines in CTIS across all scenes and in DTIS on average.

| Metric | Method | living_room_1 | living_room_2 | office_1 | office_2 | average |
|---|---|---|---|---|---|---|
| $\text{PSNR}_{part}$ ($\uparrow$) | D3DR | 9.818 | **18.072** | **15.640** | **11.857** | **11.966** |
| | LBM | 7.182 | 10.800 | 8.044 | 11.725 | 10.075 |
| | TIP-Editor | 13.259 | 7.955 | 10.019 | 0.633 | 6.960 |
| | R3DG | **13.579** | 7.810 | 7.319 | 7.280 | 8.598 |
| | Copy-Paste | 8.749 | 10.045 | 5.428 | 5.832 | 6.519 |
| | Instruct-GS2GS | 8.895 | 10.013 | 5.630 | 6.345 | 6.892 |
| $\text{PSNR}_{cropped}$ ($\uparrow$) | D3DR | 18.087 | **23.114** | **22.421** | **18.490** | **18.039** |
| | LBM | 16.025 | 16.467 | 16.050 | 17.797 | 16.271 |
| | TIP-Editor | 19.512 | 13.989 | 17.105 | 6.869 | 12.502 |
| | R3DG | **20.247** | 13.455 | 14.127 | 13.792 | 14.453 |
| | Copy-Paste | 17.415 | 15.832 | 13.541 | 12.662 | 13.032 |
| | Instruct-GS2GS | 17.525 | 15.736 | 13.718 | 13.137 | 13.360 |
| $\text{SSIM}_{cropped}$ ($\uparrow$) | D3DR | 0.630 | **0.628** | 0.727 | **0.577** | **0.640** |
| | LBM | 0.713 | 0.497 | **0.741** | 0.541 | 0.638 |
| | TIP-Editor | 0.673 | 0.390 | 0.517 | 0.268 | 0.439 |
| | R3DG | 0.643 | 0.326 | 0.362 | 0.418 | 0.449 |
| | Copy-Paste | **0.747** | 0.492 | 0.692 | 0.458 | 0.582 |
| | Instruct-GS2GS | 0.655 | 0.476 | 0.607 | 0.439 | 0.526 |
| $\text{PSNR}_{shadow}$ ($\uparrow$) | D3DR | 19.953 | **20.627** | 21.846 | **21.700** | **19.993** |
| | LBM | 21.950 | 18.909 | **23.407** | 19.318 | 19.544 |
| | TIP-Editor | 18.216 | 18.197 | 18.846 | 6.738 | 11.899 |
| | R3DG | 18.205 | 13.177 | 12.266 | 15.875 | 14.895 |
| | Copy-Paste | **22.580** | 19.117 | 23.067 | 18.983 | 19.023 |
| | Instruct-GS2GS | 21.653 | 18.063 | 21.547 | 18.479 | 17.918 |
| $\text{PSNR}_{background}$ ($\uparrow$) | D3DR | 22.622 | **21.015** | 23.998 | **25.268** | 23.383 |
| | LBM | 26.118 | 19.770 | 27.054 | 24.320 | 23.533 |
| | TIP-Editor | 22.197 | 18.906 | 22.516 | 12.325 | 16.786 |
| | R3DG | 17.231 | 13.172 | 13.462 | 16.984 | 15.224 |
| | Copy-Paste | **27.826** | 20.126 | **27.662** | 24.767 | **24.169** |
| | Instruct-GS2GS | 23.862 | 18.771 | 23.902 | 22.185 | 20.236 |
| CTIS ($\uparrow$) | D3DR | **0.636** | 0.621 | 0.641 | 0.653 | **0.646** |
| | LBM | 0.618 | 0.619 | 0.642 | 0.654 | 0.643 |
| | TIP-Editor | 0.607 | 0.605 | 0.619 | 0.616 | 0.619 |
| | R3DG | 0.613 | 0.616 | 0.639 | 0.656 | 0.639 |
| | Copy-Paste | 0.620 | **0.624** | 0.643 | 0.656 | 0.642 |
| | Instruct-GS2GS | 0.620 | 0.619 | **0.645** | **0.657** | 0.644 |
| DTIS ($\uparrow$) | D3DR | **0.543** | 0.519 | 0.513 | 0.541 | **0.529** |
| | LBM | 0.522 | 0.518 | 0.513 | 0.543 | 0.526 |
| | TIP-Editor | 0.512 | 0.501 | 0.501 | 0.515 | 0.507 |
| | R3DG | 0.520 | 0.521 | 0.517 | 0.550 | 0.527 |
| | Copy-Paste | 0.525 | **0.524** | **0.519** | **0.551** | **0.529** |
| | Instruct-GS2GS | 0.526 | 0.519 | 0.517 | 0.548 | **0.529** |

Table 6: **Synthetic Dataset full comparison 2** The table is split into two parts due to its size; this is Part 2. The first column represents metrics names, the second column represents methods names, columns 3-6 represent scenes, the last column contains averages across rows. **Bold** numbers represent the best across methods. $\uparrow$ represents that the metric is better if the value is greater. D3DR outperforms other methods on $\text{PSNR}_{part}$, $\text{PSNR}_{cropped}$, $\text{SSIM}_{cropped}$.

| Metric | Method | toaster | fabric_bin | chair | average |
|---|---|---|---|---|---|
| CTIS (↑) | D3DR | **0.647** | **0.643** | **0.640** | **0.643** |
| | LBM | 0.635 | 0.642 | 0.638 | 0.638 |
| | TIP-Editor | 0.626 | 0.620 | 0.629 | 0.625 |
| | R3DG | 0.634 | 0.603 | 0.633 | 0.623 |
| | Copy-Paste | 0.634 | 0.642 | 0.637 | 0.638 |
| | Instruct-GS2GS | 0.646 | 0.638 | 0.639 | 0.641 |
| DTIS (↑) | D3DR | 0.520 | **0.508** | 0.503 | **0.510** |
| | LBM | 0.506 | **0.508** | **0.504** | 0.506 |
| | TIP-Editor | 0.504 | 0.485 | 0.503 | 0.497 |
| | R3DG | 0.507 | 0.496 | 0.500 | 0.501 |
| | Copy-Paste | 0.505 | 0.507 | **0.504** | 0.505 |
| | Instruct-GS2GS | **0.521** | 0.504 | **0.504** | **0.510** |

Table 7: **Real Dataset full comparison.** The first column represents metrics names, the second column represents methods names, columns 3-5 represent scenes, the last column contains averages across rows. **Bold** numbers represent the best across methods. ↑ represents that the metric is better if the value is greater. D3DR consistently outperforms other methods on CTIS metrics. It also performs better on DTIS for almost every scene.

## J   More results

Fig. 17 and Fig. 18 present rendering results of different methods on the synthetic dataset. The figures demonstrate that D3DR achieves more realistic and multi-view consistent in 3DGS object insertion task compared to the baselines.

Fig. 19 shows rendering results of different methods on the real dataset. While D3DR produces more realistic object insertions than the baselines, it also fails to generate convincing shadows in most scenes. We hypothesize that this limitation arises from inaccuracies in the reconstructed camera poses, which introduce noise into the 3DGS reconstructions. As a result, the diffusion model prioritizes enhancing the overall scene appearance rather than generating shadows.

## K   Diffusion Personalization for Texture Preservation Optimization

Custom Diffusion (Kumari et al., 2023) reports faster convergence when data augmentations are applied during training. We find that cropping is essential for texture preserving diffusion model optimization to generate images with correct object orientation and texture.

During training, we apply random cropping, where the crop size increases linearly from 30% to 50% of the image over the first 500 iterations, and from 50% to 100% over iterations 500–1000. This progressive cropping strategy allows the diffusion model to make better use of the information in object images. Intuitively, in the early iterations, the model observes zoomed-in object images, so errors in texture generation carry higher loss, forcing the model to utilize the available information more effectively. Fig. 21 shows diffusion model personalization results under different cropping strategies. With a *constant crop*, the generated orientation is misaligned with the real orientation, while the *random crop* strategy produces realistic results but aligns orientation later in training compared to our approach. Additionally, the proposed approach yields the best visual texture reproduction (especially the 8th image in Fig. 21).

## L   DDS Initialization Comparison

Fig. 22 demonstrates the effectiveness of our proposed initialization for DDS optimization. Our approach produces realistic lighting at the end of optimization, whereas the vanilla DDS results in noticeable object deformation and an unrealistic appearance. The prompts for our approach are $y = a$ *cup on a plate* and $y_{\text{orig}} = a$ *plate*, while for vanilla DDS they are $y = a$ *realistic cup on a plate* and $y_{\text{orig}} = a$ *cup on a plate*.

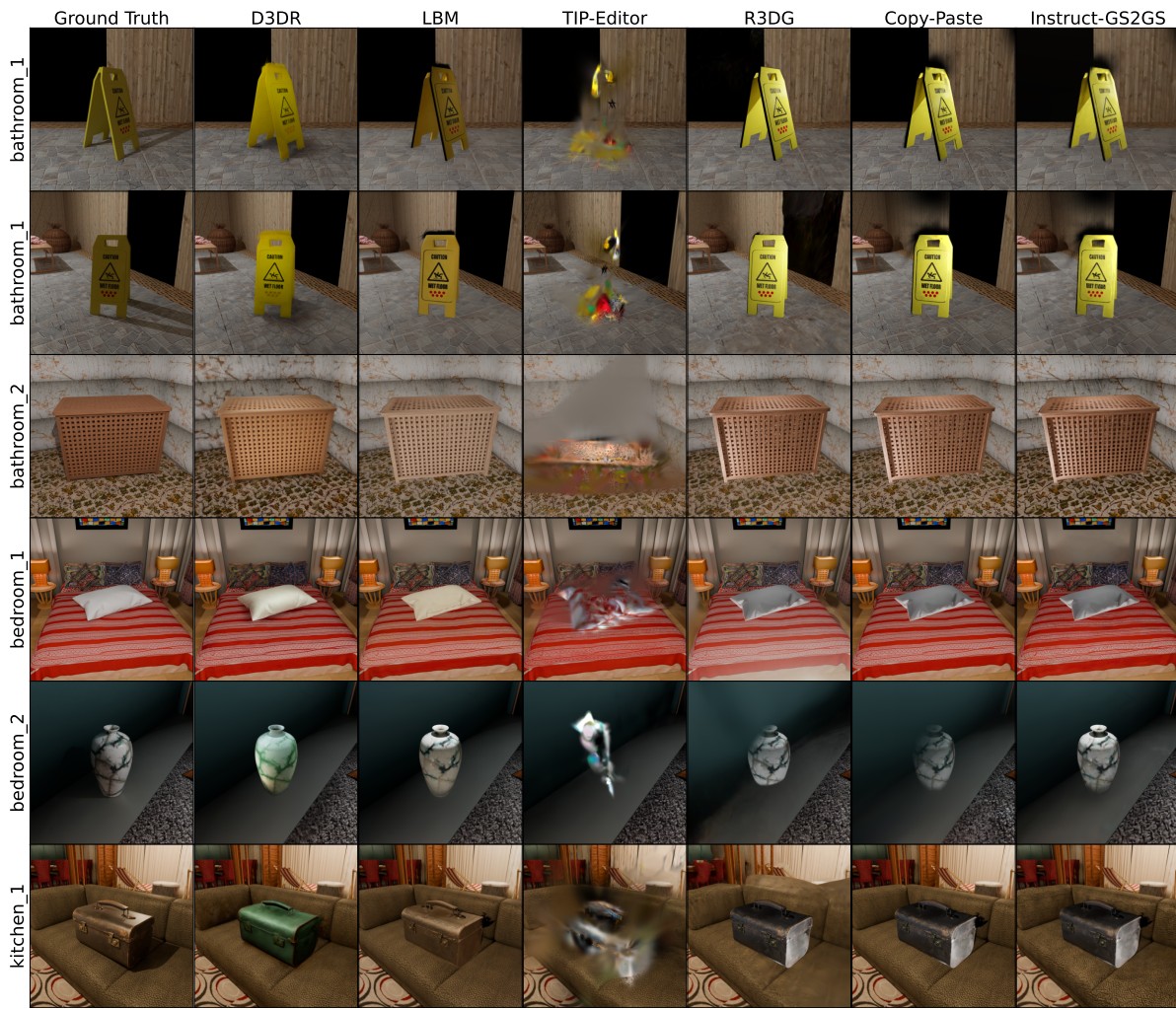

Figure 17: **Renderings of synthetic data, part 1.** Each row shows results on different scenes from the synthetic dataset. Column 1 presents the ground truth, while columns 2–6 show the results of different methods. The scene *bathroom_1* is shown twice to highlight LBM's inability to produce multi-view consistent results. TIP-Editor fails to generate large objects, and R3DG does not produce realistic object insertions. D3DR improves the appearance of inserted objects. For *kitchen_1*, however, we observe that D3DR alters the albedo, which is a common issue in diffusion-based editing approaches (Chadebec et al., 2025).

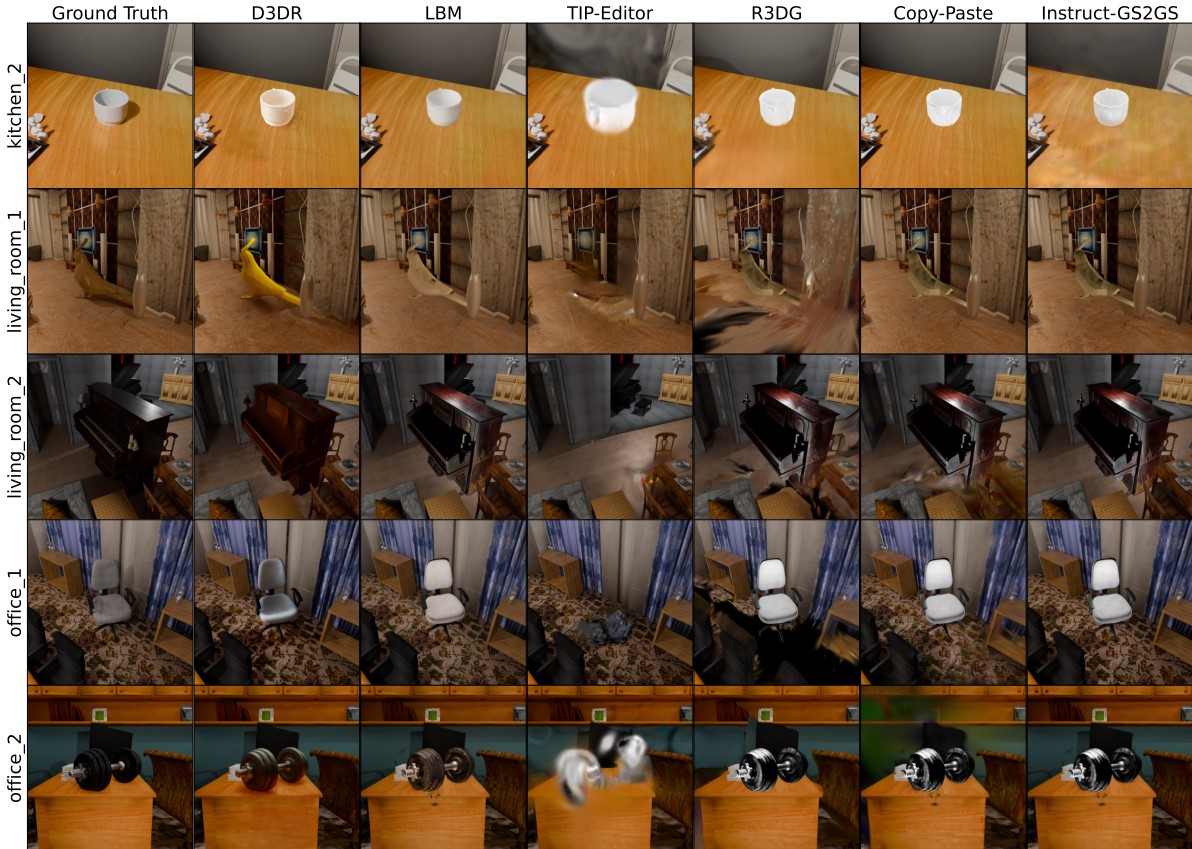

Figure 18: **Renderings of synthetic data, part 2.** Each row shows results on different scenes from the synthetic dataset. Column 1 presents the ground truth, while columns 2–6 show the results of different methods. TIP-Editor generates a small object in *kitchen_2* but struggles to generate large objects. R3DG does not produce realistic object insertions. D3DR improves the appearance of inserted objects. LBM fails to produce shadows, and its insertion results are not realistic for most scenes.

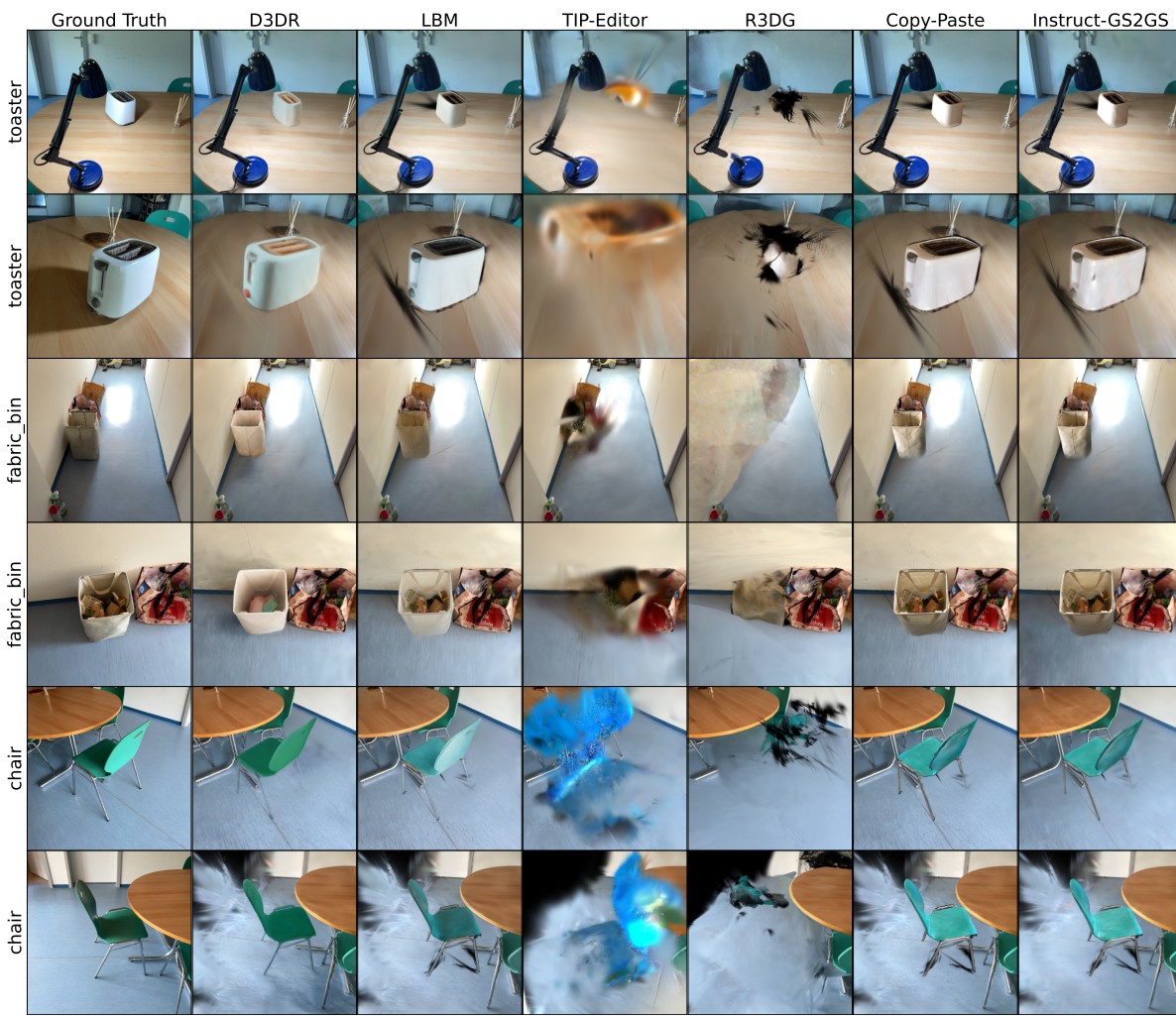

Figure 19: **Renderings of real data.** Each pair of rows shows results on different scenes from the real dataset. Column 1 presents the ground truth, while columns 2–6 show the results of different methods. TIP-Editor struggles to generate large objects or produces unrealistic ones. R3DG is not robust for real-world scenarios and fails to produce realistic object insertions. D3DR improves the appearance of inserted objects but does not generate shadows for *toaster* and *chair*. LBM also fails to produce shadows, and its results contain black artifacts.

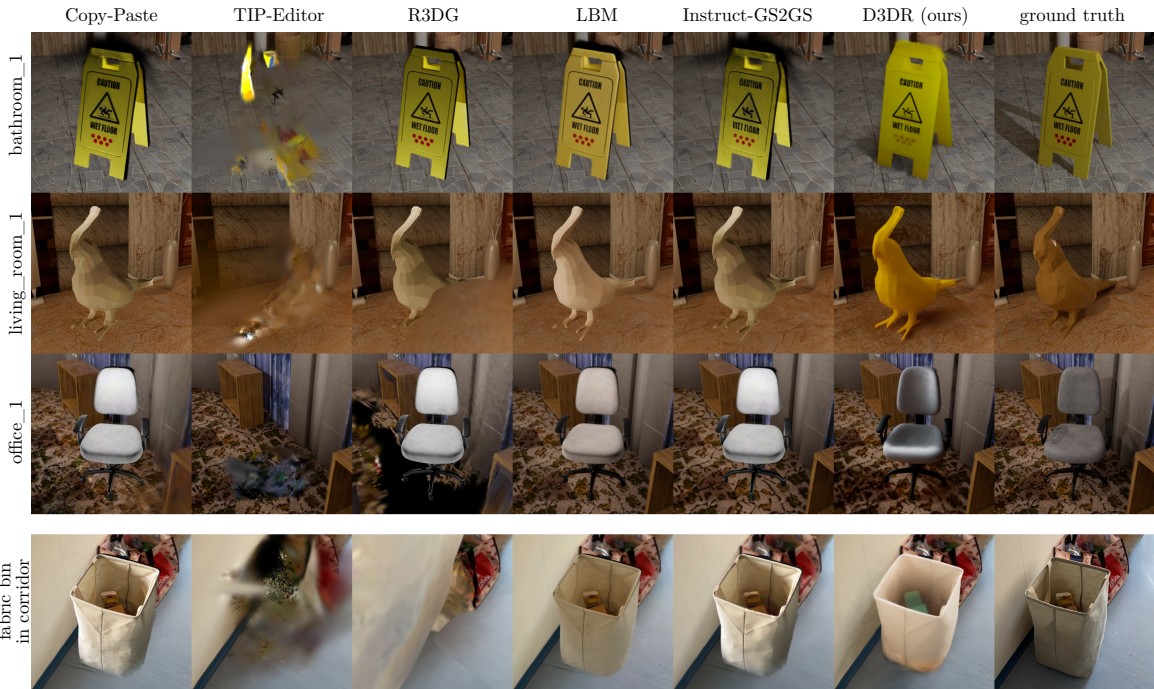

Figure 20: **Zoomed-in comparison with other methods.** Rows represent different scenes and columns represent different methods. It can be seen that our method improves object's appearance.

## M    Training Parameters

To train the rough diffusion personalization we randomly sample 32 places from a list of *"a room"*, *"a bedroom"*, *"a bathroom"*, *"a kitchen"*, *"a living room"*, *"an office"*, *"a supermarket"*, *"a restaurant"*, *"a school"*, *"a hospital"*, *"a university"*, *"a library"*, *"a classroom"*, *"a gym"*, *"a shopping mall"*, *"a movie theater"*, *"a museum"*, *"a warehouse"*, *"a factory"*, *"a laboratory"*, *"a hotel lobby"*, *"a hotel room"*, *"a conference room"*, *"a waiting room"*, *"a coffee shop"*, *"a bar"*, *"a nightclub"*, *"a TV studio"*, *"a police station"*, *"a train station"*, *"an airport terminal"*, *"a subway station"*, *"a bus terminal"*, *"a bookstore"*, *"a bank"*, *"a coworking space"*, *"a daycare center"*, *"a kindergarten"*, *"a university lecture hall"*, *"a science lab"*, *"a computer lab"*, *"a locker room"*, *"a theater stage"*, *"an art gallery"*, *"a bowling alley"*, *"a skating rink"*, *"a playroom"*, *"a cafeteria"*, *"a dining hall"*, *"a fast food restaurant"*, *"a food court"*, *"a garage"*, *"a storage unit"*, *"a utility room"*, *"a laundry room"*, *"a broom closet"*, *"a server room"*, *"a data center"*, *"a power plant control room"*, *"a control tower"*, *"an observatory"*, *"a dormitory"*, *"a monastery"*, *"a meditation room"*, *"a command center"*, *"a VIP lounge"*.

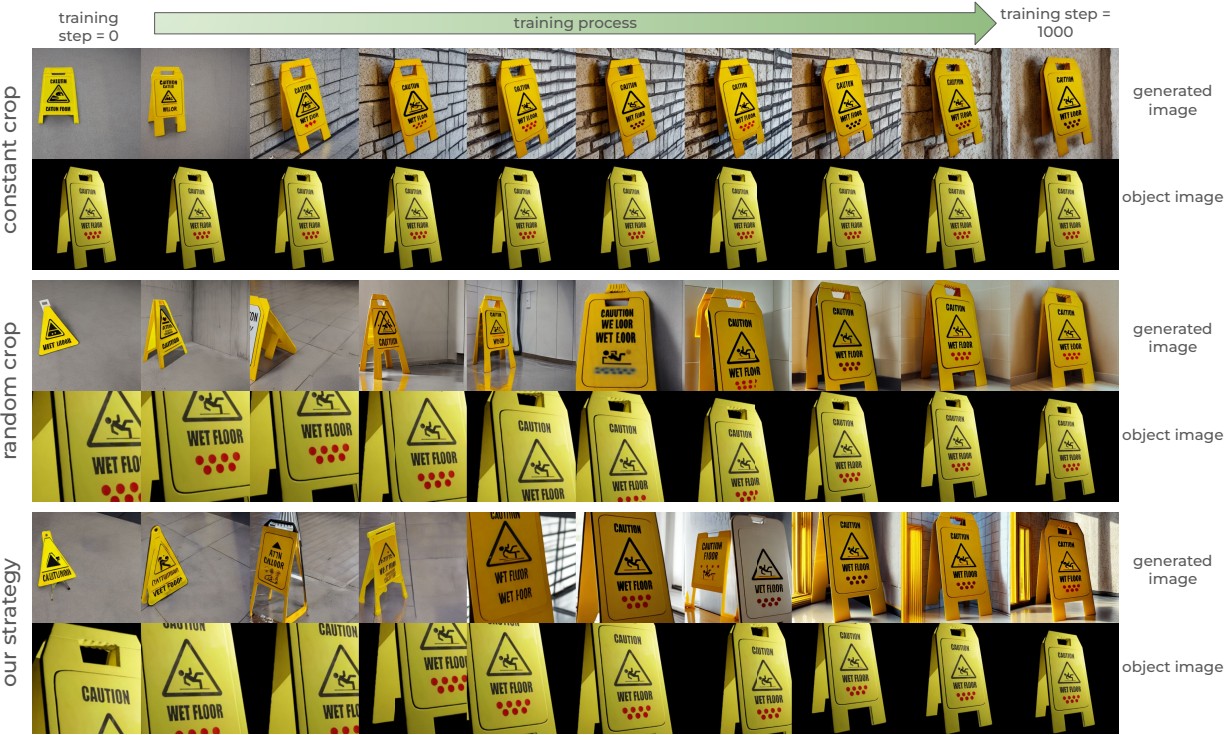

Figure 21: **Cropping strategy for texture-preserving diffusion models training.** The first two rows show diffusion model fine-tuning when the crops cover the full image. The next two rows correspond to crops with sizes randomly chosen between 30% and 100% of the image. The final row illustrates our proposed crop size strategy. Odd rows show generated images during training, while even rows display the corresponding input object images used for generation.

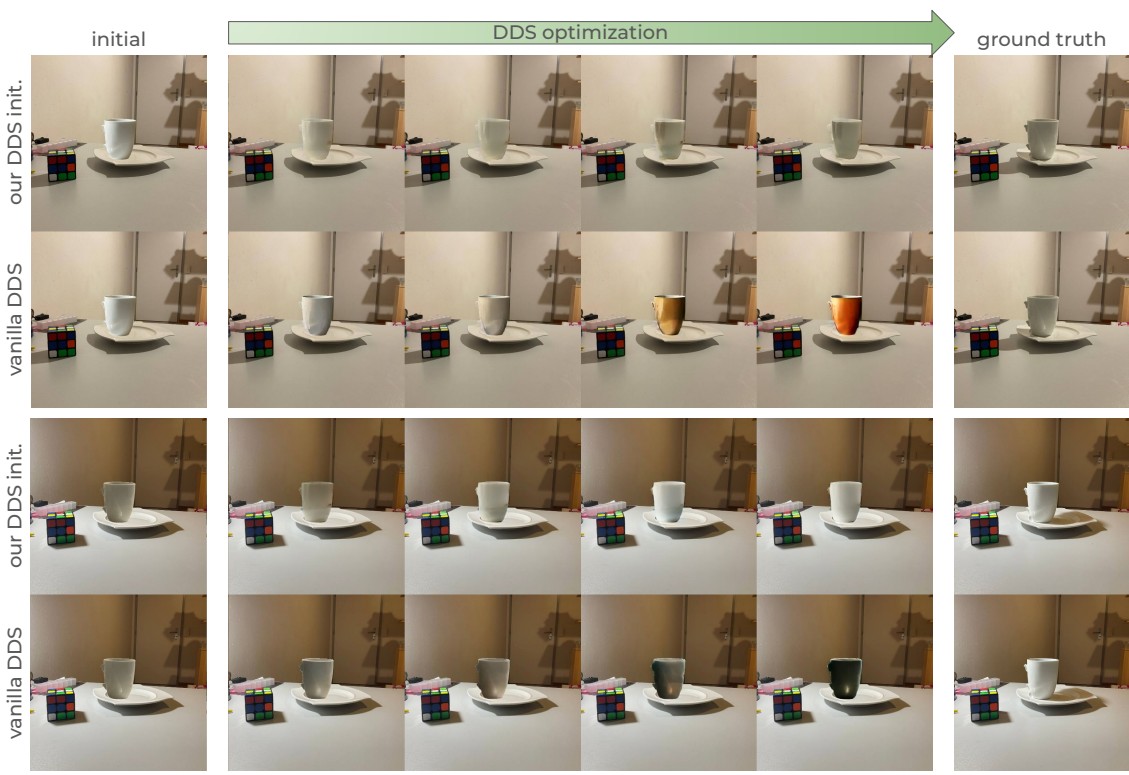

Figure 22: **DDS initialization comparison.** The first two rows and the next two rows show object insertion optimization using DDS under two different lighting conditions. Odd rows correspond to optimization with our initialization, while even rows correspond to vanilla DDS optimization.

