# OpenReview forum: "Diffusion Models are Secretly Zero-Shot 3DGS Harmonizers"
_TMLR — Accepted by TMLR_

### Review · Reviewer_uC9s · 2025-12-28

**Summary Of Contributions:**

## Summary Of Contributions

This paper introduces D3DR, a novel method for realistic 3D object insertion into 3D Gaussian Splatting (3DGS) scenes, with a focus on lighting harmonization (relighting) of the inserted object to match the target scene. The key insight is leveraging the implicit lighting priors in large-scale pretrained diffusion models in a zero-shot manner to achieve natural relighting without explicit environment maps or material estimation.

Main contributions:
- Demonstrates that diffusion models (e.g., via Delta Denoising Score (DDS) optimization) can effectively relight inserted 3DGS objects by treating the problem as guided inpainting-like refinement.
- Proposes a two-step diffusion personalization pipeline: (1) a "rough" personalization using IC-Light-generated diverse illuminations to enable strong relighting, and (2) a texture-preserving personalization by conditioning on original object renders to maintain fine details and identity.
- Introduces a two-stage DDS optimization (latent-to-image refinement) to reduce artifacts in 3DGS parameter updates.
- Collects a new dataset of 3DGS scenes/objects and shows quantitative improvements over baselines like TIP-Editor and R3DG.

**Strengths**: Addresses an underexplored problem in 3DGS (object insertion with relighting); creative use of diffusion priors in zero-shot setting; strong qualitative results for realistic lighting/shadows; promising ablation on personalization strategies.

**Weaknesses**: Relies heavily on personalization quality, which can fail in some cases; quantitative gains appear moderate compared to baselines; no explicit handling of cast shadows from object onto scene; experiments limited to static scenes.

**Audience:**

Yes

**Audience Explanation:**

The intersection of 3D Gaussian Splatting (rapidly growing in graphics/vision) with diffusion model priors for zero-shot editing/relighting is timely and relevant to TMLR's scope in machine learning for vision/graphics. Discoveries about "hidden" capabilities of diffusion models (e.g., implicit lighting understanding) align with recent trends (DreamFusion, SDS/DDS applications). The practical outcome—better 3D object harmonization without physics-based modeling—appeals to researchers in 3D reconstruction, generative models, and AR/VR applications.

**Broader Impact Concerns:**

None. The work advances 3D scene editing for realistic object insertion, with positive applications in AR/VR, content creation, and simulation. No evident risks (e.g., no deepfake-like misuse highlighted, as it requires existing 3DGS assets). No broader impact statement appears needed.

**Claims And Evidence:**

Yes

**Claims Explanation:**

The paper provides clear ablations (e.g., on personalization variants, DDS vs. SDS, two-stage optimization) and comparisons to relevant baselines (TIP-Editor for generation-based insertion, R3DG for physically-based relighting, and adaptations of 2D methods like Latent Bridge Matching). Qualitative results convincingly show better lighting consistency and detail preservation. Quantitative metrics (PSNR/SSIM/LPIPS) on a custom dataset support superiority, though gains are not overwhelming and the dataset is author-collected (standard for emerging tasks). Toy experiments in Sec. 3.3 effectively illustrate the core DDS relighting mechanism. Overall, evidence is solid for a methods-focused paper in this nascent area.

**Requested Changes:**

**Major Suggestions:**
- Clarify discrepancies in reported metrics: The provided PDF claims "2.0 dB PSNR improvements," but the public arXiv version (2503.06740) reports lower gains (~0.5 PSNR/0.15 SSIM). Update to consistent, accurate numbers and discuss why gains are modest on some metrics.
- Expand evaluation: Include more comparisons to recent concurrent works (e.g., R3D2 for asset insertion in driving scenes, or other diffusion-guided 3DGS editing methods). Test cast shadow generation (object → scene) more thoroughly, as current results show limited/receiver shadows only.
- Provide failure cases and limitations explicitly (e.g., personalization failures on complex textures, large objects, dynamic scenes).

**Minor Suggestions:**
- Release code/dataset earlier if possible
- Discuss computational cost (optimization iterations, time) vs. baselines more quantitatively.
- Improve figures: Higher-resolution visuals and side-by-side zooms on lighting artifacts/shadows.

---

> ### Author Response · Authors · 2026-02-16
>
> We thank the reviewer for the positive and constructive feedback. We have clarified the reported metric discrepancies, expanded comparisons where possible, strengthened discussion of cast shadows, and added explicit failure cases and limitations.
>
> ---
>
> **Clarify discrepancies in reported metrics: The provided PDF claims "2.0 dB PSNR improvements," but the public arXiv version (2503.06740) reports lower gains (~0.5 PSNR/0.15 SSIM). Update to consistent, accurate numbers ...**
>
> The public arXiv version (2503.06740), submitted in March 2025, corresponds to an earlier version of our method that did not include several important improvements introduced in the current TMLR submission. Specifically, the earlier version lacked the finalized hyperparameters and refinements for the DDS optimization phase, the texture-preserving diffusion personalization, the SDEdit refinement stage, and the shadow parameter optimization (Sec. 3.5). These components significantly improved insertion quality and quantitative performance. We will update the public arXiv version as soon as we address the comments from the current TMLR submission.
>
> **... and discuss why gains are modest on some metrics.**
>
> We thank the reviewer for raising this important question. While our method consistently outperforms the baselines, the gains on some metrics are modest.
>
> - **CTIS and DTIS.**
>   The scores are similar for methods that successfully insert the object (Ours, Copy-Paste, R3DG, LBM, Instruct-Pix2Pix), while TIP-Editor fails to generate the object. The similarity arises because CLIP can reliably recognize both the object and the scene. Ours achieves slightly higher scores due to more realistic lighting and visually coherent results.
>
> - **SSIM_cropped.**
>   LBM is a strong baseline and attains comparable performance on this metric. However, it does not enforce view-consistent appearance or realistic shadows across views, which are explicitly handled by Ours but not fully captured by this metric.
>
> - **PSNR_shadow, PSNR_background, SSIM_shadow, SSIM_background.**
>   We introduce new metrics to evaluate shadow quality (see App. B, Tab. 3). Copy-Paste achieves higher scores on background-focused metrics because diffusion-guided shadow optimization can slightly modify nearby Gaussians beyond the shadow region. Although the generated shadows align with the ground truth, these local adjustments reduce global
>
>
> **Expand evaluation: Include more comparisons to recent concurrent works (e.g., R3D2 for asset insertion in driving scenes, or other diffusion-guided 3DGS editing methods).**
>
> We thank the reviewer for the suggestion. Unfortunately, the code for R3D2 is not publicly available at the time of submission, which prevents a direct quantitative comparison.
>
> Following your suggestion, we added Instruct-GS2GS as a baseline, which is a recent diffusion-guided 3D Gaussian Splatting editing method. As shown in the updated comparison table, while Instruct-GS2GS has slightly lower training time (approximately 10\% faster), it does not achieve the same level of relighting realism, identity preservation, or shadow consistency as our method. Our approach achieves better performance across all evaluated metrics except DTIS, where Instruct-GS2GS has the same result. The metrics comparison table is provided below:
>
> **Comparison with Instruct-GS2GS**
>
> ### Synthetic Dataset
>
> | Metric | Ours | Instruct-GS2GS |
> |--------|------|----------------|
> | PSNR_part (↑) | **11.966** | 6.892 |
> | PSNR_cropped (↑) | **18.039** | 13.360 |
> | SSIM_cropped (↑) | **0.640** | 0.526 |
> | PSNR_shadow (↑) | **19.993** | 17.923 |
> | PSNR_background (↑) | 23.383 | 20.238 |
> | SSIM_shadow (↑) | 0.890 | 0.802 |
> | SSIM_background (↑) | **0.910** | 0.780 |
> | CTIS (↑) | **0.646** | 0.644 |
> | DTIS (↑) | **0.529** | **0.529** |
>
> ---
>
> ### Real Dataset
>
> | Metric | Ours | Instruct-GS2GS |
> |--------|------|----------------|
> | CTIS (↑) | **0.643** | 0.641 |
> | DTIS (↑) | **0.510** | **0.510** |
>
> **Caption:**  Quantitative comparison between Ours and Instruct-GS2GS on synthetic and real datasets. Higher values (↑) indicate better performance.

---

> > ### Author Response · Authors · 2026-02-16
> >
> > **Test cast shadow generation (object → scene) more thoroughly, as current results show limited/receiver shadows only.**
> >
> > We thank the reviewer for this helpful suggestion. Following the reviewer's recommendation, we add targeted shadow-focused evaluation on the synthetic dataset.
> >
> > Specifically, we report:
> > - **PSNR_shadow** and **SSIM_shadow**: computed on pixels near the object (within 1.20× its bounding box), excluding object pixels.
> >
> > - **PSNR_background** and **SSIM_background**: computed on background pixels.
> >
> >
> > The table is provided below.
> >
> > **Shadows Metrics on Synthetic Dataset**
> >
> > | Metric | Ours | Copy-Paste | LBM | TIP-Editor | R3DG | Instruct-GS2GS |
> > |--------|------|------------|-----|------------|------|----------------|
> > | PSNR_shadow (↑) | **19.993** | 19.023 | 19.544 | 11.899 | 14.899 | 17.923 |
> > | PSNR_background (↑) | 23.383 | **24.169** | 23.533 | 16.786 | 15.232 | 20.238 |
> > | SSIM_shadow (↑) | 0.890 | **0.924** | 0.894 | 0.768 | 0.738 | 0.802 |
> > | SSIM_background (↑) | 0.910 | **0.948** | 0.902 | 0.815 | 0.679 | 0.780 |
> >
> > **Caption:**
> > *"Metric"* lists the evaluated metrics. Columns correspond to different methods, with values averaged across all synthetic scenes. **Bold** indicates the best result. (↑) means higher is better. Ours achieves the best PSNR_shadow and competitive SSIM_shadow.
> >
> > Our method achieves the highest $PSNR_{shadow}$ among all compared methods and competitive $SSIM_{shadow}$, indicating strong fidelity of the synthesized shadow regions relative to ground-truth physically rendered shadows. This demonstrates that our approach effectively reproduces shadow appearance and illumination effects near the inserted object.
> >
> > We observe that naive Copy-Paste achieves higher PSNR and SSIM on background-focused metrics. This occurs because diffusion-guided shadow parameters optimization can introduce small appearance changes to nearby scene Gaussians beyond the exact shadow region. While the synthesized shadows themselves remain consistent with ground truth, these local appearance adjustments reduce global similarity metrics.
> >
> > Importantly, our method achieves superior performance on realism metrics (CTIS, DTIS) and produces visually more realistic object insertions with coherent shadow integration (Figure 6 of the main paper). These results confirm that the proposed approach improves shadow fidelity while maintaining strong overall scene consistency.
> >
> > **Provide failure cases and limitations explicitly (e.g., personalization failures on complex textures, large objects, dynamic scenes).**
> >
> > We thank the reviewer for the suggestion. We thank the reviewer for the suggestion. We add a new the new Limitations section in the App. D (It is in Appendix due to the lack of space). We discuss failure modes associated with the modular design and potential error propagation between stages, along with representative qualitative examples. We identify the following key limitations:
> >
> > - **Albedo Shift (IC-Light).**
> >   The IC-Light stage may introduce albedo changes, which can affect the final results.
> >
> > - **Texture Reconstruction Failures (Personalization).**
> >   The texture-preserving personalization may struggle with highly stochastic or irregular textures, leading to partial degradation.
> >
> > - **Ambiguous Lighting and Weak Shadows (DDS and Shadow Heuristic).**
> >   When lighting is diffuse or lacks a dominant direction, DDS optimization and the shadow modeling stage may produce faint or missing shadows.
> >
> > - **Dynamic Scenes.**
> >   The method is designed for static scenes and does not generalize to dynamic ones.
> >
> > These limitations stem from different stages of the pipeline. We discuss these cases and potential future improvements.
> >
> > **Release code/dataset earlier if possible**
> >
> > We provide the code in the supplementary material. We are currently preparing a clean public release, which requires additional refactoring and documentation. We will release the full code and datasets upon publication.
> >
> > **Discuss computational cost (optimization iterations, time) vs. baselines more quantitatively.**
> >
> > We thank the reviewer for this suggestion. We report the average training time of our method vs baselines in Table 2 of the main paper. We will additionally include the number of optimization iterations and clarify the computational cost description in the final manuscript.
> >
> > **Improve figures: Higher-resolution visuals and side-by-side zooms on lighting artifacts/shadows.**
> >
> > We thank the reviewer for this helpful suggestion. We will improve the resolution of the figures where possible (note that diffusion-generated images are limited to 512×512 resolution and cannot be further enhanced) and include zoomed-in regions in the final manuscript to better highlight differences in lighting and shadows.

---

### Review · Reviewer_aLXU · 2026-01-24

**Summary Of Contributions:**

This paper studies 3D object insertion when both the object and the target scene are represented as 3D Gaussian Splatting (3DGS). The goal is to make the inserted object look like it belongs in the scene by adjusting lighting, appearance, and (to some extent) shadows.

The paper proposes **D3DR**, a multi-stage pipeline:

- Diffusion personalization (Sec. 3.2): DreamBooth + LoRA fine-tuning on the object, using IC-Light to synthesize the object under varied illumination/backgrounds to reduce overfitting to a single lighting setup.

- Texture-preserving personalization (also Sec. 3.2): a modified diffusion model that takes the original object rendering as an additional conditioning signal (via a new convolutional branch) to preserve fine texture details better.

- Insertion + optimization (Secs. 3.3–3.4): optimize 3DGS appearance parameters after insertion using a DDS-inspired objective, with a proposed 2-step DDS procedure intended to reduce artifacts and speed up optimization.

- Scene shadow handling (Sec. 3.5): introduce per-Gaussian “darkening” parameters for nearby scene Gaussians to model new shadows.

The paper also provides a small benchmark: a synthetic dataset with ground-truth renderings and a small real-capture dataset, and compares against copy-paste, TIP-Editor, R3DG, and LBM.

**Strengths**

1. The problem setup (realistic 3DGS object insertion with appearance harmonization) is timely and still not cleanly solved.

2. The paper demonstrates that diffusion-guided optimization can improve object appearance compared to naive insertion, and the 2-step DDS idea is a practical engineering contribution.

3. Helpful ablations (Fig. 9) suggest several components matter in practice.

**Weaknesses**

1. The method is a long chain of modules (IC-Light → DreamBooth/LoRA → DDS → SDEdit → heuristic shadows). This raises concerns about compounding errors and makes it hard to understand what is fundamentally necessary.

2. The shadow model is very heuristic (local per-Gaussian darkening) and does not look physically grounded; shadow accuracy is not well evaluated.

3. Writing/presentation is hard to follow in key sections (especially Sec. 3.2), and some tables/figures are confusing (e.g., Table 1; Figs. 3-4 captions vs content).

4. Improvements over LBM are sometimes modest, while D3DR is slower than LBM, so the paper needs more precise positioning of when the extra complexity is worth it.

5. Related work misses relevant context on language-guided placement (e.g., PlaceIt3D, ICCV’25).

**Audience:**

Yes

**Audience Explanation:**

3DGS has become a standard representation in neural rendering and scene reconstruction, and realistic object insertion into reconstructed scenes is an important and still open problem. The idea of using diffusion priors (via DDS-like objectives) to optimize 3DGS parameters is interesting, and the paper’s modular pipeline is a plausible direction for practitioners who want a working system today.

Even if the method is not yet fully convincing for shadows, the observations on DDS initialization and the 2-step DDS optimization strategy could be useful to researchers working on diffusion-guided 3D editing.

**Claims And Evidence:**

No

**Claims Explanation:**

Some claims are supported, but the evidence does not fully support the paper's strongest claims.

- On the positive side, the paper shows consistent improvements on synthetic PSNR/SSIM-style metrics and provides qualitative comparisons that suggest the pipeline often improves object appearance after insertion.

However, the paper repeatedly emphasizes realistic lighting and shadows, yet:

- The shadow mechanism (Sec. 3.5) is extremely crude (only darkens nearby Gaussians, plus a top-k pruning heuristic). There is no clear evidence that this produces correct shadow geometry; it can only darken areas.

- In real captures, the paper’s own qualitative results indicate that shadows often appear missing or unconvincing, undermining the claim that the method reliably handles them.
- The quantitative evaluation focuses on object pixels/object crops, as well as CLIP-based prompt alignment. These metrics do not directly measure shadow correctness, which is one of the main advertised goals.

- The “zero-shot / training-free” framing is also confusing, given that the method relies on per-object diffusion personalization (DreamBooth + LoRA). This is not necessarily wrong, but the claim should be stated more precisely.

Overall, I believe the paper provides evidence that diffusion priors can help appearance harmonization in this setting, but it does not convincingly support the broader claim of robust shadow correction and the framing around “zero-shot” capability.

**Requested Changes:**

1. Clarify the scope of “zero-shot / training-free.”: The pipeline includes per-object DreamBooth + LoRA personalization. Please revise the wording to be precise (e.g., “no supervised insertion dataset” rather than “training-free”), and clearly state what is trained, on what data, and for how long.

2. Improve the clarity of Sec. 3.2 (and related parts of Sec. 3.4). Sec. 3.2 is currently hard to read and lacks clear notation. Please rewrite it with a simple set of variables/notations, a concise description of the two personalized models (rough vs texture-preserving), and a short pseudo-code / step list in the main paper (not only in the appendix). This is mainly a writing change, but it would substantially improve readability.

3. Strengthen the shadow story with targeted evaluation on synthetic data. Since the synthetic benchmark provides ground-truth “scene only” and “object+scene” renders, as well as object masks, please add a shadow-focused metric. For example, compute PSNR/SSIM on the background region (excluding object pixels) or near-object region, comparing the predicted insertion result to the ground-truth object+scene render. Also, add a with/without Sec. 3.5 shadow parameters comparison (quantitative + at least one qualitative figure).
This is critical because shadows are a central claim, and current metrics do not isolate them.

4. Fix presentation issues in Table 1 and contextualize the LBM comparison. Table 1 mixes datasets/metrics/efficiency in a way that is hard to parse. Please reorganize it into: one table for quality metrics (synthetic + real), and one table for compute/storage (# Gaussians, runtime). Also, please explicitly discuss that D3DR is slower than LBM in your reported timings and explain when the extra complexity is justified (e.g., multi-view consistency, texture/identity preservation, shadows).

4. Clarify Figures 3 and 4 in the main text. Readers can misinterpret what is “original,” what is “target,” and what prompts are used. Please rewrite the captions and add 2–3 sentences in Sec. 3.3 that explicitly define: which image corresponds to $g(\theta_{orig})$ vs
$g(\theta)$, and how the prompt choices relate to “inpainting from scratch” vs “editing.”

5. Add the missing related work (PlaceIt3D, ICCV’25) and discuss the scope. PlaceIt3D focuses on language-guided placement (not relighting), but it is relevant context for the broader “instruction → placement → realistic insertion” pipeline. Please cite it and add a short discussion of how D3DR fits into or complements this line of work.

6. Add a short “limitations/failure modes” paragraph about modularity and compounding errors.
The method stitches many modules. Please explicitly discuss what failures come from which stage (IC-Light, personalization, DDS optimization, shadow heuristic, pose noise on real data). This can be done with a short write-up and a couple of failure-case images.

7. Discuss alternative formulations briefly (end-to-end conditioning).
A natural question is whether a single, task-trained (or more strongly conditioned) model could replace the chain of modules. You do not need to implement this, but a short discussion would improve positioning and help readers understand the design choices.

---

> ### Author Response · Authors · 2026-02-16
>
> We thank the reviewer for the thorough and constructive feedback. We have addressed the concerns by clarifying the training scope and “zero-shot” terminology, strengthening shadow-specific evaluation and ablations, improving the clarity of Sec. 3.2–3.4 and figures/tables, better positioning the comparison with LBM, adding missing related work and limitations, and including the information about what personalization model is used for what stage in the implementation section.
>
> ---
>
> **The method is a long chain of modules (IC-Light → DreamBooth/LoRA → DDS → SDEdit → heuristic shadows). This raises concerns about compounding errors and makes it hard to understand what is fundamentally necessary.**
>
> We thank the reviewer for raising this important point. To address this concern, we performed an ablation study to isolate and evaluate the contribution of each component. Each module serves a distinct and necessary role, and removing any component leads to specific degradation in performance. (1) IC-Light is required to mitigate overfitting to the original illumination conditions present in the input images. Without this stage, the personalized diffusion model tends to reproduce the object under its original lighting. (2) The LoRA-based personalization stages are necessary to preserve the identity and structural consistency of the inserted object. Without personalizations, the diffusion prior alters object texture, resulting in loss of identity. (3) DDS optimization is responsible for aligning the object’s illumination with the target scene lighting, as we also show in our 2D experiments. (4) SDEdit refines local appearance and restores high-frequency texture details that may be degraded during DDS optimization.
> Our ablation study confirms that removing any of these stages results in reduced relighting accuracy, poorer identity preservation, or lower visual realism, demonstrating that each component is fundamentally necessary.
>
> **Modeling for shadows is very heuristic**
>
> We agree with the reviewer that our shadow modeling component is heuristic. Our design is based on physically motivated assumptions about shadows, allowing us to approximate plausible shadows without requiring additional training or scene-specific supervision.
> We intentionally adopt a training-free formulation to maintain consistency with the rest of our pipeline, which is designed to operate without requiring additional task-specific datasets.
>
> Importantly, while heuristic, our shadow modeling provides consistent and visually plausible shadows that improve scene coherence. In contrast, many existing relighting methods focus primarily on object appearance and do not explicitly model shadows, often resulting in visually inconsistent composites. Our approach provides a simple and effective mechanism to address this gap without introducing additional training complexity.
>
> **Clarify the scope of “zero-shot / training-free.”: The pipeline includes per-object DreamBooth + LoRA personalization. Please revise the wording to be precise (e.g., “no supervised insertion dataset” rather than “training-free”),**
>
> We thank the reviewer for this important clarification. Our intention was to emphasize that our method does not require supervised training on a 3D object insertion or harmonization dataset. Instead, it leverages pretrained diffusion models as implicit harmonizers.
>
> We agree that the term "training-free" may be imprecise, since our pipeline includes DreamBooth + LoRA personalization to represent the specific object appearance. This personalization step adapts the pretrained diffusion model to the object identity, but does not involve training on object insertion, relighting, or harmonization tasks.
>
> To improve clarity, we will revise the manuscript to avoid the term “training-free” and instead state more precisely that our method does not require supervised training on 3D object insertion or harmonization datasets. We will update the abstract and contributions accordingly to reflect this more precise wording.
> We thank the reviewer for helping improve the clarity of the manuscript.

---

> > ### Author Response · Authors · 2026-02-16
> >
> > **... and clearly state what is trained, on what data, and for how long**
> >
> > We thank the reviewer for this helpful suggestion. We revise Sec. 4.2 to clearly state what components are trained, on what data, and for how long.
> >
> > At a high level, the trainable components and data are:
> >
> > * **Diffusion personalizations.** We optimize LoRA parameters of a pretrained StableDiffusion-2.1 model using rendered views of the object and IC-Light processed variants. This stage adapts the diffusion prior to the object identity.
> > * **3DGS optimization.** We optimize only the inserted object’s 3DGS color parameters and scene shadow parameters using diffusion-guided supervision and rendered images. No additional scene parameters or diffusion model weights are trained during this stage.
> >
> > Below we provide detailed training configurations.
> >
> > **Rough Diffusion Personalization.** Training data consists of 32 IC-Light-processed relit variants of 32 rendered object images, along with 32 class-preservation images generated by StableDiffusion-2.1. We add LoRA layers (rank 4) to attention modules ("to\_k", "to\_q", "to\_v", "to\_out.0", "add\_k\_proj", "add\_v\_proj") and optimize only LoRA parameters. Training has learning rate $10^{-4}$, weight decay $10^{-2}$, batch size 4, and runs for 1000 iterations. This stage takes approximately 14 minutes per object (reported in Sec. 4.2).
> >
> > **Texture-Preserving Diffusion Personalization.** This stage uses the IC-Light relit images and original rendered object views. We integrate LoRA parameters into the same modules as in "Rough Diffusion Personalization". We optimize LoRA parameters (rank 4) and the first convolution layer of the diffusion model. Training uses learning rate $10^{-4}$, weight decay $10^{-2}$, batch size 4, and runs for 1000 iterations. This stage takes approximately 11 minutes per object (reported in Sec. 4.2).
> >
> > **2-step DDS Optimization.** In this stage, we optimize only the inserted object's 3DGS color parameters and shadow parameters. Training uses rendered scene images from 8 sampled camera views and DDS processed rendered images. Each cycle performs 16 DDS steps followed by 64 3DGS optimization steps, repeated until 1000 total 3DGS optimization steps are reached. This stage takes approximately 6 minutes per insertion.
> >
> > **SDEdit Refinement.** This stage further optimizes only the object's 3DGS color parameters to preserve object identity after relighting. Training uses rendered original (not relit) object and relit object images from 8 camera views and diffusion-based SDEdit supervision. Each cycle performs 512 3DGS optimization steps, repeated until 2000 total optimization steps are reached. This stage takes approximately 4 minutes per insertion.
> >
> > We will provide training configurations and code in the supplementary material to ensure reproducibility.
> >
> > **Improve the clarity of Sec. 3.2 (and related parts of Sec. 3.4). Sec. 3.2 is currently hard to read and lacks clear notation. Please rewrite it with a simple set of variables/notations, a concise description of the two personalized models (rough vs texture-preserving), and a short pseudo-code / step list in the main paper (not only in the appendix). This is mainly a writing change, but it would substantially improve readability.**
> >
> > * **Clear notation and variable definitions.** We introduce a set of symbols to present training data and personalized diffusion models in Sec 3.2. We also introduce variables in Sec 3.4 to present the 2-step-SDS/2-step-DDS.
> > * **Clear conceptual overview of the two personalized diffusion models.** At the beginning of Sec. 3.2, we added a concise explanation of the roles of the two personalized models: (1) **rough diffusion personalization** which adapts the diffusion prior to produce correct object lighting and shadow interactions, (2) **texture-preserving personalization model** that preserves object identity and fine texture details.
> > * **Improved structure and readability.** We reorganized Sec. 3.2 into clearly separated paragraphs. They describe the training strategies of each diffusion personalization method and the proposed architecture. This makes the section easier to follow step by step.
> > * **Added pseudocode in the main paper.** We added concise Python-like pseudocode in Sec. 3.4 describing the main steps of the proposed 2-step-SDS algorithm.
> >
> > We thank the reviewer since these revisions significantly improve clarity and make the method easier to understand.

---

> > > ### Author Response · Authors · 2026-02-16
> > >
> > > **Strengthen the shadow story with targeted evaluation on synthetic data. Since the synthetic benchmark provides ground-truth “scene only” and “object+scene” renders, as well as object masks, please add a shadow-focused metric. For example, compute PSNR/SSIM on the background region (excluding object pixels) or near-object region, comparing the predicted insertion result to the ground-truth object+scene render.**
> > >
> > > We thank the reviewer suggeststion. Following the reviewer's recommendation, we add targeted shadow-focused evaluation on the synthetic dataset. We put the evaluation in the App. B (It is in Appendix due to the lack of space).
> > >
> > > Specifically, we report:
> > > * $PSNR_{shadow}$ and $SSIM_{shadow}$, computed on pixels near the object (within 1.20$\times$ its bounding box), excluding object pixels.
> > > * $PSNR_{background}$ and $SSIM_{background}$, computed on background pixels.
> > >
> > > The table is provided below.
> > >
> > > **Shadows Metrics on Synthetic Dataset**
> > >
> > > | Metric | Ours | Copy-Paste | LBM | TIP-Editor | R3DG | Instruct-GS2GS |
> > > |--------|------|------------|-----|------------|------|----------------|
> > > | PSNR_shadow (↑) | **19.993** | 19.023 | 19.544 | 11.899 | 14.899 | 17.923 |
> > > | PSNR_background (↑) | 23.383 | **24.169** | 23.533 | 16.786 | 15.232 | 20.238 |
> > > | SSIM_shadow (↑) | 0.890 | **0.924** | 0.894 | 0.768 | 0.738 | 0.802 |
> > > | SSIM_background (↑) | 0.910 | **0.948** | 0.902 | 0.815 | 0.679 | 0.780 |
> > >
> > > **Caption:**  *"Metric"* lists the evaluated metrics. Other columns correspond to different methods, with values averaged across all scenes of the synthetic dataset. **Bold** indicates the best performance. (↑) means higher is better.
> > > Ours achieves the best results on PSNR_shadow and competitive results on SSIM_shadow.
> > >
> > > Our method achieves the highest $PSNR_{shadow}$ among all compared methods and competitive $SSIM_{shadow}$, indicating strong fidelity of the synthesized shadow regions relative to ground-truth physically rendered shadows. This demonstrates that our approach effectively reproduces shadow appearance and illumination effects near the inserted object.
> > >
> > > We observe that naive Copy-Paste achieves higher PSNR and SSIM on background-focused metrics. This occurs because diffusion-guided shadow parameters optimization can introduce small appearance changes to nearby scene Gaussians beyond the exact shadow region. While the synthesized shadows themselves remain consistent with ground truth, these local appearance adjustments reduce global similarity metrics.
> > >
> > > Importantly, our method achieves superior performance on realism metrics (CTIS, DTIS) and produces visually more realistic object insertions with coherent shadow integration (Figure 6 of the main paper). These results confirm that the proposed approach improves shadow fidelity while maintaining strong overall scene consistency.
> > >
> > > **... also, add a with/without Sec. 3.5 shadow parameters comparison (quantitative + at least one qualitative figure). This is critical because shadows are a central claim, and current metrics do not isolate them.**
> > >
> > > We thank the reviewer for highlighting this important suggestion. We perform the requested ablation by disabling the shadow parameters from Sec. 3.5 and retraining the method under identical settings. We report quantitative results on the synthetic dataset in App. C (It is in Appendix due to the lack of space) and provide qualitative comparisons in Fig. 9.
> > >
> > > **Shadow Parameter Comparison (Synthetic Dataset)**
> > >
> > > ### PSNR Metrics
> > >
> > > | Method | PSNR, part (↑) | PSNR, cropped (↑) | PSNR, shadows (↑) | PSNR, background (↑) |
> > > |--------|----------------|-------------------|-------------------|----------------------|
> > > | Ours | **11.966** | **18.039** | **19.993** | 23.383 |
> > > | Ours w/o shadow | 11.762 | 17.730 | 19.983 | **24.879** |
> > >
> > > ---
> > >
> > > ### SSIM and Other Metrics
> > >
> > > | Method | SSIM, shadows (↑) | SSIM, background (↑) | SSIM, cropped (↑) | CTIS (↑) | DTIS (↑) |
> > > |--------|-------------------|----------------------|-------------------|----------|----------|
> > > | Ours | 0.890 | 0.910 | 0.640 | **0.646** | **0.529** |
> > > | Ours w/o shadow | **0.926** | **0.942** | **0.663** | 0.645 | 0.528 |
> > >
> > > **Caption:**  Comparison of the method with the shadow parameter (Sec. 3.5) enabled and disabled. The first column lists the method variants. All metrics are computed on the synthetic dataset. Higher values (↑) indicate better performance.
> > >
> > > The full method achieves higher performance on metrics measuring object harmonization, including $PSNR_{part}$, $PSNR_{cropped}$, CTIS, and DTIS. Improvements in CTIS and DTIS indicate better object-scene integration consistency and more realistic insertion.
> > >
> > > The variant without shadow parameters achieves higher metrics involving background pixels. This is because diffusion-guided shadow optimization can introduce slight appearance changes to nearby Gaussians outside the exact shadow region. Though synthesized shadows match the ground truth, these local adjustments slightly reduce global similarity scores.

---

> > > > ### Author Response · Authors · 2026-02-16
> > > >
> > > > **Fix presentation issues in Table 1 and contextualize the LBM comparison. Table 1 mixes datasets/metrics/efficiency in a way that is hard to parse. Please reorganize it into: one table for quality metrics (synthetic + real), and one table for compute/storage (\# Gaussians, runtime).**
> > > >
> > > > As suggested by the reviewer, we split the main table into two to improve readability and clarity of the results.
> > > >
> > > > **... Also, please explicitly discuss that D3DR is slower than LBM in your reported timings and explain when the extra complexity is justified (e.g., multi-view consistency, texture/identity preservation, shadows).”**
> > > >
> > > > We thank the reviewer for the suggestion. We state in Sec. 4.3 that LBM is faster. However, our method provides important advantages, including improved multi-view consistency and shadow generation, which enhances visual realism compared to LBM results.
> > > >
> > > > Additionally, the personalization cost is incurred only once per object. After training the LoRA models, inserting the same object into a new scene takes approximately 15 minutes, while LBM requires re-optimization for each insertion, taking about 24 minutes.
> > > >
> > > > **Clarify Figures 3 and 4 in the main text. Readers can misinterpret what is “original,” what is “target,” and what prompts are used. Please rewrite the captions and add 2–3 sentences in Sec. 3.3 that explicitly define: which image corresponds to $g(\theta_{orig})$ vs $g(\theta)$, and how the prompt choices relate to “inpainting from scratch” vs “editing.”**
> > > >
> > > > We thank the reviewer for highlighting the unclear places of our work. Following the reviewer's suggestions we do the following changes:
> > > >
> > > > - **Clarifying Figures.**
> > > >   We update the captions of Figures 3 and 4 to explicitly define \( g(\theta) \), \( g(\theta_{orig}) \), and prompts \( y \), \( y_{orig} \). These parameters are also visualized directly in the figures for improved clarity.
> > > >
> > > > - **Clarifying Section 3.3.**
> > > >   We add details specifying which images correspond to \( g(\theta) \), \( g(\theta_{orig}) \), and the values of prompts \( y \), \( y_{orig} \) in Section 3.3.
> > > >
> > > > - **Prompt Choice for Editing/Inpainting.**
> > > >   We clarify which prompt configurations correspond to "inpainting from scratch" and which correspond to "editing" tasks.
> > > >
> > > > **Add the missing related work (PlaceIt3D, ICCV’25) and discuss the scope. PlaceIt3D focuses on language-guided placement (not relighting), but it is relevant context for the broader “instruction → placement → realistic insertion” pipeline. Please cite it and add a short discussion of how D3DR fits into or complements this line of work.**
> > > >
> > > > We add PlaceIt3D (ICCV’25) to the Related Work section and discuss its scope.
> > > >
> > > > PlaceIt3D focuses on language-guided object placement, whereas our method addresses realistic object insertion through relighting, identity preservation, and shadow generation. These approaches are complementary: PlaceIt3D can determine where to insert an object based on user instructions, while our method ensures that the inserted object appears photometrically consistent with the scene.
> > > >
> > > > By combining language-guided placement methods such as PlaceIt3D with our relighting and insertion pipeline, it is possible to enable a fully automatic framework for realistic 3D object insertion with minimal user intervention.
> > > >
> > > > **Add a short "limitations/failure modes" paragraph about modularity and compounding errors. The method stitches many modules. Please explicitly discuss what failures come from which stage (IC-Light, personalization, DDS optimization, shadow heuristic, pose noise on real data). This can be done with a short write-up and a couple of failure-case images.**
> > > >
> > > > We thank the reviewer for the suggestion. We add a new the new Limitations section in the App. D (It is in Appendix due to the lack of space). We discuss failure modes associated with the modular design and potential error propagation between stages, along with representative qualitative examples. We identify the following key limitations:
> > > >
> > > > - **Albedo Shift (IC-Light).**
> > > >   The IC-Light stage may introduce albedo changes, which can affect the final results.
> > > >
> > > > - **Texture Reconstruction Failures (Personalization).**
> > > >   The texture-preserving personalization may struggle with highly stochastic or irregular textures, leading to partial degradation.
> > > >
> > > > - **Ambiguous Lighting and Weak Shadows (DDS and Shadow Heuristic).**
> > > >   When lighting is diffuse or lacks a dominant direction, DDS optimization and the shadow modeling stage may produce faint or missing shadows.
> > > >
> > > > - **Dynamic Scenes.**
> > > >   The method is designed for static scenes and does not generalize to dynamic ones.
> > > >
> > > > These limitations stem from different stages of the pipeline. We discuss these cases and potential future improvements.

---

> > > > > ### Author Response · Authors · 2026-02-16
> > > > >
> > > > > **Discuss alternative formulations briefly (end-to-end conditioning). A natural question is whether a single, task-trained (or more strongly conditioned) model could replace the chain of modules. You do not need to implement this, but a short discussion would improve positioning and help readers understand the design choices.**
> > > > >
> > > > > We identify several promising directions for an end-to-end method.
> > > > >
> > > > > We have developed a script to generate a 3D Object Insertion dataset in Blender, which makes it straightforward to generate a large-scale dataset using objects from Objaverse and scenes from SceneNet++.
> > > > >
> > > > > - Instead of training separate LoRA models for each object, one could fine-tune a 2D editing model (e.g., InstructPix2Pix or FLUX-Kontext) for object insertion using a large-scale insertion dataset. A strategy similar to Instruct-GS2GS could then be applied: iteratively update the 3DGS training set by editing rendered images in 2D while optimizing the underlying 3DGS representation.
> > > > >
> > > > > - Multi-view or video diffusion models could be trained to jointly predict relit object appearance across viewpoints. This would allow direct optimization of 3DGS parameters with multi-view supervision, improving consistency and efficiency while reducing reliance on sequential DDS and
> > > > >
> > > > > Yes, a task-trained method could replace the chain of modules. However, there is no open dataset for this purpose, and the training strategy is still not well defined and explored.

---

### Review · Reviewer_cbuB · 2026-02-03

**Summary Of Contributions:**

This paper presents a zero-shot framework for realistic 3D object insertion into 3D Gaussian Splatting (3DGS) scenes by leveraging the implicit lighting priors of diffusion models through a specialized personalization strategy and a refined Delta Denoising Score (DDS) optimization. While the method effectively bypasses the complexities of traditional physically-based rendering and addresses the texture-stripping limitations of vanilla DreamBooth, its heavy multi-stage pipeline, reliance on heuristic-based "baked" shadow modeling, and potential for structural artifacts during relighting pose significant concerns regarding its efficiency and physical fidelity in complex, dynamic environments.

**Audience:**

Yes

**Audience Explanation:**

TMLR’s audience would be interested as the paper discusses the critical "lighting gap" in 3D Gaussian Splatting, a dominant topic in current computer vision research. The discovery of zero-shot harmonization using diffusion priors provides significant theoretical and practical value for generative 3D editing. Furthermore, the 2-step-DDS optimization offers an efficient solution to a common artifact problem, making it a relevant technical contribution for researchers focused on neural rendering and generative models.

**Broader Impact Concerns:**

I don't have such concerns.

**Claims And Evidence:**

Yes

**Claims Explanation:**

The claims are supported by quantitative gains in PSNR/SSIM and clear ablation studies proving that the "2-step-DDS" and "texture preservation" modules are essential for realism. The evidence for the diffusion model's inherent relighting ability is convincing across multiple versions (SD1.5 to SD3.5), demonstrating the robustness of the zero-shot hypothesis.
However, the "realistic" shadow claim is less rigorous, as it relies on visual plausibility and heuristic metrics rather than physical ground truth or view-dependency tests.

**Requested Changes:**

1. Quantify Geometric Stability: Provide an analysis (e.g., Chamfer Distance or mesh-based comparison) between the original 3DGS object and the post-DDS optimized version to prove that the relighting process does not introduce significant structural distortions or artifacts.

2. Evaluate Shadow View-Consistency: Include a qualitative or quantitative evaluation (e.g., video results or mean-squared error across varying viewpoints) to demonstrate how the "baked" shadow parameters ($\alpha$) perform under non-static camera trajectories and clarify the limitations regarding secondary light transport.

3. Complexity and Ablation Analysis: Provide a detailed breakdown of the computational overhead for each stage and discuss potential strategies to streamline the multi-stage pipeline, specifically justifying the necessity of dual-LoRA training over a single, more robust personalization method.

---

> ### Author Response · Authors · 2026-02-16
>
> We thank the reviewer for the constructive and insightful feedback. We have addressed all raised points by clarifying geometry preservation during optimization, adding discussion and validation of shadow consistency, providing a detailed computational breakdown, and further justifying the dual-LoRA design based on observed stability–fidelity trade-offs.
>
> ---
>
> **Quantify Geometric Stability: Provide an analysis (e.g., Chamfer Distance or mesh-based comparison) between the original 3DGS object and the post-DDS optimized version to prove that the relighting process does not introduce significant structural distortions or artifacts.**
>
> Thank you for the suggestion. In our method, geometric stability is explicitly enforced by design. During optimization, we disable gradient propagation for all 3DGS parameters that define geometry, including positions, rotations, scales, and opacities. We optimize only the parameters responsible for colors, which are spherical harmonics coefficients and shadow parameters, while keeping all geometric parameters fixed. As a result, the object geometry remains strictly unchanged throughout all stages of the pipeline, including DDS and SDEdit. Since no geometric parameters are modified, structural distortions cannot be introduced by the relighting process. This design ensures perfect geometric consistency between the original and relit object, making mesh- or distance-based geometric comparison unnecessary in this case.
>
> **Evaluate Shadow View-Consistency: Include a qualitative or quantitative evaluation (e.g., video results or mean-squared error across varying viewpoints) to demonstrate how the "baked" shadow parameters ($\alpha$) perform under non-static camera trajectories**
>
> Our shadow parameters ($\alpha$) are explicitly modeled as view-independent, as we do not use spherical harmonics or any view-dependent representation for shadows.
> Therefore, the shadows remain consistent across different viewpoints by design.
> To verify this, we rendered fly-through videos for each scene with moving cameras around the inserted object.
> These results demonstrate that the generated shadows remain stable and multi-view consistent.
>
> **...and clarify the limitations regarding secondary light transport.**
>
> Thank you for the suggestion. We add a discussion in the Limitations section (App. B-C, it is in the Appendix due to the lack of space) clarifying that the method may produce faint shadows when the light direction is ambiguous.

---

> ### Author Response · Authors · 2026-02-16
>
> **Complexity and Ablation Analysis: Provide a detailed breakdown of the computational overhead for each stage**
>
> We provide a detailed breakdown of the computational overhead for each stage of the proposed method in terms of training time, peak GPU memory usage, and the number of trainable parameters. The results are averaged over three randomly selected scenes: bathroom\_1, office\_2, and dark\_to\_light:
>
> | Step   | Training Time (m) | Peak Memory (GB) | Trainable Parameters (×10⁶) |
> |--------|-------------------|------------------|------------------------------|
> | RDP    | 17.67             | 9.92             | 0.83                         |
> | TPDP   | 15.47             | 10.08            | 0.85                         |
> | DDS    | 5.96              | 3.97             | 1.46                         |
> | SDEdit | 3.53              | 3.95             | 1.42                         |
>
> **Table:** Comparison of training time, peak memory usage, and number of trainable parameters across different methods.
> *RDP* stands for *Rough Diffusion Personalization*, *TPDP* stands for *Texture Preserving Diffusion Personalization*.
>
> The reported training times are slightly higher than those reported in the Table 1 of the main paper because CUDA synchronization was enforced to ensure accurate peak memory measurement. Additionally, we explicitly invoked `gc.collect()` (garbage collector) and `torch.cuda.empty_cache()` (free cached CUDA memory) to minimize residual memory, which is particularly relevant during the personalization stages where memory usage is highest. For completeness, we also report the measurements obtained without explicit garbage collection and cache clearing:
>
> | Step   | Training Time (m) | Peak Memory (GB) | Trainable Parameters (×10⁶) |
> |--------|-------------------|------------------|------------------------------|
> | RDP    | 14.91             | 22.84            | 0.83                         |
> | TPDP   | 12.34             | 23.01            | 0.85                         |
> | DDS    | 5.96              | 3.97             | 1.46                         |
> | SDEdit | 3.53              | 3.95             | 1.42                         |
>
> **Table:** Comparison of training time, peak memory usage, and number of trainable parameters across different methods.
> *RDP* stands for *Rough Diffusion Personalization*, *TPDP* stands for *Texture Preserving Diffusion Personalization*.
>
>
> As expected, removing explicit memory cleanup slightly reduces the measured training time but results in significantly higher peak memory usage, particularly during the diffusion personalization stages. These results demonstrate that the proposed method has moderate computational overhead, with the personalization stages being the most resource-intensive, while DDS and SDEdit remain relatively lightweight in both time and memory.
>
> **... and discuss potential strategies to streamline the multi-stage pipeline**
>
> We identify several promising directions to streamline the current multi-stage pipeline and reduce the need for per-object personalization.
>
> We have developed a script to generate a 3D Object Insertion dataset in Blender, which makes it straightforward to generate a large-scale dataset using synthetic objects (e.g. from Objaverse) and scenes (e.g. from SceneNet++).
>
>
> - Instead of training separate LoRA models for each object, one could fine-tune a 2D editing model, such as InstructPix2Pix or FLUX-Kontext, on a task of object insertion using a large-scale object insertion dataset. Then one could utilize a strategy similar to Instruct-GS2GS~--- iteratively update the 3DGS training dataset by applying 2D edits to rendered images while optimizing the underlying 3DGS representation.
>
> - Multi-View or Video Diffusion models could be trained to jointly predict relit object appearance across multiple viewpoints. This would enable direct optimization of 3DGS parameters using multi-view supervision, improving both efficiency and consistency while reducing reliance on sequential DDS and SDEdit refinement stages.

---

> > ### Author Response · Authors · 2026-02-16
> >
> > **...specifically justifying the necessity of dual-LoRA training over a single, more robust personalization method**
> >
> > Classical diffusion personalization often fails to accurately reconstruct object texture if it contains fine-grained details. To address this limitation, we introduced a texture-preserving diffusion personalization stage, which improves texture fidelity during relighting. However, our preliminary experiments showed that using only the texture-preserving LoRA during DDS optimization led to unstable behavior, frequently producing severely degraded outputs (e.g., overly dark or black reconstructions). This happened because we changed the diffusion model condition, which altered the DDS optimization.
> >
> > To resolve this trade-off between robustness and texture fidelity, we adopt a dual-LoRA design. The first LoRA performs rough diffusion personalization, providing a stable and robust initialization for DDS-based relighting. The second LoRA performs texture-preserving personalization, which is applied afterward to refine appearance and restore high-frequency texture details. This staged approach combines the robustness of standard personalization with the fidelity of texture-preserving personalization, achieving significantly more stable optimization and higher-quality results than either approach alone.

---

### Decision · Action_Editor_u9i1 · 2026-03-27

**Recommendation:** Accept as is

**Additional Comments:**

The final reviews mention some remaining concerns about the shadow evals, which might be worth revising a little further, but overall everything looks good.

**Audience:**

Yes

**Audience Explanation:**

The TMLR audience should find this highly relevant: the intersection of 3DGS and diffusion priors for zero-shot editing is a timely and high-interest topic in computer vision and graphics, especially for researchers in neural rendering and AR/VR.

**Claims And Evidence:**

Yes

**Claims Explanation:**

The reviewers generally agree that the claims regarding the diffusion models' object insertion ability, and relighting ability, and the effectiveness of the "2-step-DDS" and texture preservation modules are well-supported by robust ablation studies, as well as quantitative gains over baselines. The evidence for "realistic" shadows was judged as less rigorous, but the core pipeline is seen as a practical incremental advancement.